# Identification of Key Genes Associated with Endoplasmic Reticulum Stress in Calcium Oxalate Kidney Stones

**DOI:** 10.3390/genes16111338

**Published:** 2025-11-06

**Authors:** Zhenkun Tan, Wusheng She, Boqiang Wang, Xiang Wang, Xiaofeng Guan, Zhiwei Tao, Yaoliang Deng

**Affiliations:** 1Department of Urology, The Second Affiliated Hospital, Guangxi Medical University, Nanning 530000, China; 202310251@sr.gxmu.edu.cn (Z.T.);; 2Department of Urology, The First Affiliated Hospital, Guangxi Medical University, Nanning 530000, China

**Keywords:** calcium oxalate kidney stones, endoplasmic reticulum stress, key gene, nomogram

## Abstract

Background: Previous studies have indicated an association between endoplasmic reticulum stress (ERS) and the formation of kidney stones. To further investigate this mechanism, this research sought to identify key genes linked to ERS in calcium oxalate (CaOx) kidney stones. Methods: Key cells with the highest ERS-related gene (ERSRG) scores were identified through single-cell analysis. These key cells were then categorized into high- and low-score groups based on their average ERSRG scores. To identify key genes, we analyzed the intersection of key ERSRGs and differentially expressed genes (DEGs) within key cells, focusing on genes demonstrating significant expression differences between control and CaOx kidney stone samples. A nomogram was constructed using these key genes to predict the risk of CaOx kidney stones. Gene set enrichment analysis (GSEA) was further performed to explore the functions of these key genes in the disease. Additionally, secondary clustering analysis was conducted on key cells to identify subtypes and evaluate the expression of key genes within these subtypes. Finally, the identified key genes were validated using quantitative real-time PCR (qRT-PCR) and Western blot analysis on cultured HK-2 cells, which were exposed with 2 mM CaOx for 24 h at 37 °C with 5% CO_2_ or incubated with regular culture medium. Results: Endothelial cells were identified as key cells, and nine key genes were pinpointed in CaOx kidney stones: *ACSL4*, *PTK2*, *DUSP4*, *MMP7*, *PHLDB2*, *TGM2*, *PPT1*, *SPARCL1*, and *LTF*. The nomogram developed from these key genes demonstrated robust predictive ability for CaOx kidney stones risk. Additionally, GSEA revealed that olfactory transduction was enriched by key genes except *PTK2*. Secondary clustering analysis identified four key cell subtypes within endothelial cells, with *LTF*, *MMP7*, and *SPARCL1* showing significantly differential expression between control and CaOx kidney stones groups across all key cell subtypes. qRT-PCR and Western blot analyses revealed that, compared to the control group, CaOx-exposed HK-2 cells exhibited significantly increased expression of *ACSL4*, *MMP7*, *TGM2*, *PPT1*, and *LTF* (*p* < 0.05), while showing significantly decreased expression of *PTK2*, *DUSP4*, *SPARCL1*, and *PHLDB2* (*p* < 0.05). Conclusions: This study identified key genes associated with ERS in CaOx kidney stones through single-cell and transcriptomic analysis. The discovery of these genes provides new insights into the treatment of CaOx kidney stones and offers valuable references for subsequent research. Future research should focus on elucidating the precise roles of these candidate genes in CaOx stone pathogenesis to assess their potential for therapeutic intervention.

## 1. Introduction

Nephrolithiasis, a common urological disorder affecting approximately 14.8% of individuals globally [1], results from the aggregation of mineral and salt crystals within the renal system [2]. This condition is marked by its frequent occurrence, propensity for recurrence, and substantial healthcare expenditures, collectively diminishing patient well-being [3]. The development of kidney stones is often linked to elevated levels of calcium, oxalate, and uric acid in urine [4]. When the concentrations of these substances surpass their solubility limits in urine, supersaturation ensues, leading to the formation of urinary crystals [5]. Calcium oxalate (CaOx) kidney stones are the predominant type, characterized by a calcium content of at least 75% within the stone [6]. Research indicates that the formation of kidney stones is influenced by environmental, metabolic, and genetic factors, but the critical factor is the damage to kidney tubular epithelial cells caused by abnormal urinary components, particularly hyperoxaluria, which plays a pivotal role in stone formation [7]. Thus, identifying key genes in CaOx kidney stones is crucial for uncovering potential therapeutic targets.

Endoplasmic reticulum (ER) is a vital organelle in eukaryotic cells, responsible for maintaining cellular homeostasis through protein, lipid, and steroid synthesis, as well as calcium-dependent signaling [8]. ERS is induced by the build-up of misfolded proteins within the ER [9]. In response, the unfolded protein response (UPR) is activated to enhance protein folding within the ER [10]. Notably, our research group has previously established a foundational mechanistic link between ERS and CaOx nephrolithiasis. We demonstrated that excessive activation of ERS has been linked to the development of CaOx kidney stones. Continuous exposure of kidney tubular epithelial cells to oxalate or CaOx can lead to hyperactivation of the UPR, resulting in cellular damage [11]. Moreover, inhibiting excessive autophagy through ERS pathways can offer renoprotection by mitigating apoptosis and preventing kidney stone development [12]. These observations highlight the critical role of ERS in the pathogenesis and progression of CaOx kidney stones.

However, while these prior studies confirmed the importance of ERS at a generalized tissue and pathway level, they focused on predefined signaling cascades such as PERK-eIF2α. To precisely delineate the cellular and genomic architecture of ERS in CaOx nephrolithiasis, the present study first sought to pinpoint the key cell types exhibiting the most pronounced ERS signature, subsequently identifying a refined set of ERS-related key genes by integrating single-cell RNA sequencing (scRNA-seq) and bulk transcriptomics. This strategy allows us to shift from asking “whether” ERS is important to identifying “which specific cells and genes” are the principal actors within the complex renal microenvironment in human CaOx stone disease. Through correlation analysis, differential expression, and HK-2 cells validation analyses leveraging these genes, we constructed a prognostic nomogram to evaluate their clinical predictive capacity and performed functional enrichment analyses to elucidate their underlying mechanisms. Furthermore, we conducted secondary clustering of the key cells to resolve the expression patterns of these genes at a subpopulation level. Collectively, these findings are designed to provide a foundational resource and novel insights for identifying candidate key targets regulating CaOx kidney stones.

## 2. Materials and Methods

### 2.1. Data Source

CaOx kidney stones-related scRNA-seq dataset GSE176155 and transcriptome dataset GSE73680 were both retrieved from the Gene Expression Omnibus (GEO) database (http://www.ncbi.nlm.nih.gov/geo/, accessed on 8 June 2025). The GSE176155 dataset, sequenced using the GPL24676 platform, was composed of 3 Randall’s plaques and 3 normal kidney papillae tissues. The GSE73680 dataset was composed of 29 samples of kidney tissue containing CaOx kidney stones and 33 control samples. The samples were obtained from renal papillary tissue. Samples were collected via nephrectomy (mucosal excision) or PCNL/RIRS (biopsy forceps). The total RNA was subsequently extracted [13]. Furthermore, 551 ERSRGs were obtained from the GeneCards database (https://www.genecards.org/, accessed on 18 June 2025) with a relevance score of ≥10 (Appendix A).

### 2.2. ScRNA-Seq Analysis

In the GSE176155 dataset, quality control was conducted using the “seurat” package (version 4.1.0) [14] to remove unsuitable cells. The filtering criteria included: eliminating cells with fewer than 200 and greater than 4000 genes, and genes detected in fewer than 3 cells. Genes with an ncount between 200 and 20,000 were retained. Subsequently, the vst method in FindVariableFeatures function was used to extract the top 2000 highly variable genes. ScaleData function was applied to scale single-cell data, and the top 30 principle components (PCs) were extracted using JackStrawPlot and principle component analysis (PCA). Furthermore, FindNeighbors and FindClusters functions from the “seurat” package were employed to identify cell clusters, with results visualized by uniform manifold approximation and projection (UMAP). Cell subtypes were then annotated using marker genes from the CellMarker database (http://117.50.127.228/CellMarker/, accessed on 10 June 2025). To minimize the impact of doublets, “DoubletFinder” package (version 2.0.4) [15] was used to remove doublets in single-cell data.

To reliably quantify ER stress response activity at the single-cell level, we employed a consensus approach. ERSRGs scores for cell subtypes were calculated using AUCell, UCell, singscore, ssGSEA, and AddModule Score algorithms [16] and were averaged to create a composite score, thereby minimizing the limitations of any single method. The cell subtype with the highest mean composite score was identified as the key responder. Furthermore, to investigate the functional heterogeneity within this key subtype, we partitioned its cells into high and low ERSRG score groups, using the median composite score as a natural cutoff to define the two distinct states. Differential expression analysis was processed to identify DEGs between different score group with |avg log_2_FC| > 0.5 and *p* < 0.05 [17]. Spearman analysis was conducted between genes in key cells and ERSRGs scores, and key ERSRGs were selected based on |cor| > 0.1 and *p* < 0.05. Biological functions and signaling pathways associated with DEGs and key ERSRGs were investigated through Gene Ontology (GO) and Kyoto Encyclopedia of Genes and Genomes (KEGG) enrichment analyses. These analyses were conducted using the “clusterProfiler” package (version 4.2.2) [18] with *p* < 0.05.

The “Scissor” package (version 2.0.0) [19] was used to identify disease-associated cell subpopulations by integrating the scRNA-seq data (GSE176155) with bulk transcriptomic phenotypes (GSE73680). The method correlates single-cell expression profiles with bulk sample phenotypes across common genes, employing quantile normalization to mitigate batch effects and a network-regularized sparse regression model to select high-confidence cells [20]. With parameters set to alpha = 0.05, cutoff = 0.2, and family = “binomial”, Scissor^+^ cells (positively correlated with CaOx stones) and Scissor^−^ cells (negatively correlated) were identified based on their association strength with the disease phenotype. To investigate the communication between different cell subtypes, the “CellPhoneDB” package (version 1.6.1, https://github.com/sqjin/CellChat, accessed on 15 June 2025) [21] was employed to analyze the interaction number and weight among these subtypes.

### 2.3. Identification of Key Genes

To identify candidate genes, the overlapping genes between DEGs and key ERSRGs, identified using the “VennDiagram” package (version 1.2.2) [22], were selected. The expression of candidate genes was then assessed in the GSE73680 dataset using the Wilcoxon test. Genes with *p* < 0.05 were selected as key genes for further analysis.

Gene sequences of the key genes were obtained from the National Center for Biotechnology Information (NCBI) database (https://www.ncbi.nlm.nih.gov/gene, accessed on 15 June 2025), and their subcellular locations were predicted using the mRNALocater database. mRNALocater was utilized to predict mRNA (not protein) localization, given its high-throughput efficiency and reliability in eukaryotes. The resulting predictions—spanning the cytoplasm, endoplasmic reticulum, extracellular region, mitochondria, and nucleus—facilitated subsequent functional annotation and analysis of post-transcriptional regulatory mechanisms [23]. Further Pearson analysis was performed to explore the correlations among key genes. The GeneMANIA database was used to identify genes that functionally interacted with key genes and to elucidate their corresponding functions. To further investigate the molecular functions and mechanisms of key genes, GO and KEGG enrichment analyses were conducted with *p* < 0.05.

### 2.4. Construction of Nomogram

To predict the risk of CaOx kidney stones, a nomogram was developed using the expression levels of key genes. Within this nomogram, a score was assigned to each gene, with the sum of these scores representing the total points, which correlated positively with disease risk. The performance of the nomogram was subsequently assessed using calibration curves, receiver operating characteristic (ROC) curves, and decision curve analysis (DCA).

### 2.5. Gene Set Enrichment Analysis (GSEA)

To investigate the biological functions of key genes in CaOx kidney stones, GSEA was performed using “clusterProfiler” package (*p* < 0.05). The correlations between key genes with all genes were calculated and ranked. Background gene set “c2.cp.kegg.v7.4.symbols.gmt”, used in this analysis, was downloaded from the MSigDB.

### 2.6. Drug Prediction and Molecular Docking

Potential therapeutic drugs for CaOx kidney stones were identified using the DGIdb database (https://dgidb.org/, accessed on 25 June 2025). The key gene–drug network was then constructed and visualized by “Cytoscape” software (version 3.8.2) [24]. Drugs with the highest interaction scores and available 3D structure were selected for molecular docking. The 3D structures (SDF format) of selected drugs were downloaded from the PubChem database (https://pubchem.ncbi.nlm.nih.gov, accessed on 26 June 2025), and converted to PDB format by Babel GUI. The 3D protein structures of key genes were obtained from Protein Data Bank database (http://www.rcsb.org/pdb/, accessed on 28 June 2025). These protein structures were prepared by removing water molecules and small molecules using PyMOL software (version 2.5.2), and molecular docking simulations were then conducted with AutoDock Vina (version 1.1.2) [25].

### 2.7. Regulatory Network and Drug Prediction of Key Genes

To elucidate regulatory mechanisms of key genes, microRNAs (miRNAs) targeted key genes (target score ≥ 90) and long noncoding RNAs (lncRNAs) (clipExpNum > 10) targeted miRNAs were, respectively, predicted by miRDB (https://mirdb.org/, accessed on 7 July 2025) and starBase databases (https://rnasysu.com/encori/, accessed on 8 July 2025). Transcription factors (TFs) regulated key genes were predicted by the ChEA3 database (https://maayanlab.cloud/chea3/, accessed on 10 July 2025). The key gene-miRNA-lncRNA and key gene-TF regulatory networks were visualized by “Cytoscape” software.

### 2.8. Pseudotime Analysis and Secondary Clustering

The “Monocle2” package (version 2.26.0) [26] was employed to conduct pseudotime analysis on key cells to evaluate the expression change in key genes within key cells differentiation. Based on identified key cells, secondary clustering analysis was conducted for the annotation of key cell subtypes. The proportion of key cell subtypes in both control and CaOx kidney stones groups was then examined. Finally, the expression of key genes in key cell subtypes was explored.

### 2.9. Cell Culture

HK-2 cells and their culture medium were purchased from Pro-cell (CL-0109, Wuhan, China). Cells were then maintained at 37 °C with 5% CO_2_. Subsequently, the HK-2 cells were stimulated for 24 h with either 2 mM calcium oxalate (289841, CaOx, Sigma-Aldrich, St. Louis, MO, USA) or an equal volume of normal culture medium. Three culture flasks per group (control and CaOx) were harvested and used as independent biological replicates.

### 2.10. Quantitative RT-PCR

After cell culture and harvesting, total RNA was isolated from HK-2 cells using a commercially available RNA extraction kit (R401-01, Vazyme, Nanjing, China). RNA concentration was quantified spectrophotometrically (NanoDrop 2000, Thermo Fisher Scientific, Waltham, MA, USA). cDNA synthesis was performed with 1 μg total RNA using the Takara PrimeScript™ RT Master Mix (RR092A, TaKaRa, Dalian, China). qRT-PCR was conducted in 10 μL reactions containing 1 μL cDNA and SYBR Green qPCR mix (CN830A, Takara, Dalian, China), amplified on a Gentier 96 Real-Time PCR System (Tianlong, Xi’an, China). The 2^−ΔΔCT^ method was employed to determine gene expression, involving the analysis of three technical replicates per sample across three independent experiments. All primers were custom synthesized by Sangon Biotech (Shanghai, China), and their respective sequences are listed in Table 1.

### 2.11. Western Blot

HK-2 cells were harvested and then lysed using RIPA buffer, supplemented with 1% (*v*/*v*) PMSF (R0010, Solarbio, Beijing, China; G2008-1ML, Servicebio, Wuhan, China). Protein concentrations were measured using a BCA assay kit (ZJ101, EpiZyme, Shanghai, China). Equal amounts of protein samples (30 µg per lane) were then loaded onto 4–12.5% SDS-PAGE gels for electrophoretic separation. Subsequently, resolved proteins were transferred to polyvinylidene fluoride (PVDF) membranes (ISEQ00010, 0.22 µm pore size; Millipore, Burlington, MA, USA). Membranes were blocked with 5% non-fat milk for 2 h at room temperature, followed by incubation with appropriate primary antibodies overnight at 4 °C. The next day, membranes were incubated with corresponding horseradish peroxidase (HRP)-conjugated secondary antibodies for 1 h at room temperature. Primary antibodies against the following targets were used: *ACSL4* (DY1198, Abways, Shanghai, China), *PTK2* (ET1602-25, HUABIO, Hangzhou, China), *MMP7* (HA723043, HUABIO, Hangzhou, China), *TGM2* (ET1706-35, HUABIO, Hangzhou, China), *LTF* (ET7109-95, HUABIO, Hangzhou, China), *DUSP4* (YT6141, Immunoway, San Jose, CA, USA), *SPARCL1* (YN2049, Immunoway, San Jose, CA, USA), *PHLDB2* (27940-1-AP, Proteintech, Wuhan, China), and *PPT1* (RPG621Mu01, Cloud-Clone, Wuhan, China). Goat anti-rabbit IgG secondary antibody was obtained from Abways (AB0101, Shanghai, China). Protein bands were detected using a Tianneng chemiluminescence imaging system (Tianneng, Shanghai, China).

### 2.12. Statistical Analysis

All bioinformatics analyses were performed using R software (version 4.5). Differences between groups were assessed using the Wilcoxon test. Comparisons of mRNA and protein expression between the control and CaOx groups were assessed using Student’s *t*-test. Statistical significance was determined using GraphPad Prism 10 (GraphPad Software, San Diego, CA, USA), with *p* < 0.05 representing significance.

## 3. Results

### 3.1. Endothelial Cells Were Selected as Key Cells in Caox Kidney Stones

To achieve a more comprehensive understanding of the cellular signature associated with CaOx kidney stones, single-cell analysis was employed on the GSE176155 dataset. After filtering single-cell data, 27,238 cells and 28,876 genes were retained for subsequent analysis (Appendix A). The top 30 PCs were extracted (Appendix A). PCA revealed a discernible separation between the control and CaOx kidney stone (Case) groups, indicating distinct global transcriptomic profiles at the single-cell level (Appendix A). Further, twelve cell clusters were identified (Figure 1A). These clusters were annotated into nine cell subtypes (Figure 1B,C), including principal cells, T cells, fibroblasts, endothelial cells, Loop of Henle cells, macrophages, proximal tubule cells, immune cells (B cells/plasma cells), and renal vesicle cells. Among these cell subtypes, principal cells were the most abundant in control samples, while T cells were more prevalent in CaOx kidney stones samples (Figure 1D). After removing 2043 doublets, 25,195 cells were retained for further analysis (Figure 1E,F).

The ERSRGs score in these cell subtypes was calculated using AUCell, UCell, singscore, ssGSEA, and AddModule Score algorithms. Endothelial cells had the highest average scores and were identified as key cells. This computational finding was visually supported by UMAP visualization, which revealed distinct expression landscape alterations in endothelial cells from CaOx stone samples compared to the controls (Appendix A), suggesting potential disease-associated functional changes. However, these observations remain computationally derived and require further experimental validation. These endothelial cells were further classified into high-scoring and low-scoring groups based on their average ERSRGs score (Figure 2A,B). Spearman correlation analysis was conducted between all genes and the average ERSRGs scores in endothelial cells, identifying 3483 key ERSRGs with |cor| > 0.1 and *p* < 0.05 (Appendix A).

Functional annotations were performed using an unadjusted *p* < 0.05 threshold, a strategy appropriate for our exploratory study in the novel field of CaOx kidney stone ERS, as it reduces the risk of Type II errors and highlights potential mechanisms for future validation. Consequently, we identified 2612 significantly enriched GO terms and 155 KEGG pathways from the key ERSRGs. Among these items, GO terms included proteasomal protein catabolic process, focal adhesion, and cadherin binding (Figure 2C), while KEGG pathways contained protein processing in endoplasmic reticulum, pathways of neurodegeneration-multiple diseases, and endocytosis (Figure 2D) (Appendix A). Furthermore, differential expression analysis was then conducted to compare the high-scoring and low-scoring groups of endothelial cells, 851 DEGs were screened, with 689 upregulated and 162 downregulated genes (Appendix A). Enrichment analysis of DEGs identified 2141 GO terms and 105 KEGG pathways (Figure 2E,F). The enriched GO terms consisted of wound healing, focal adhesion, and cadherin binding, while KEGG pathways included fluid shear stress and atherosclerosis, cell adhesion molecules, and viral myocarditis (Appendix A).

Further analysis using the Scissor algorithm, which links single-cell phenotypes to bulk transcriptomic profiles of the disease, revealed distinct cell-type-specific associations with CaOx kidney stones. The identification of both Scissor^+^ and Scissor^−^ cells within principal cells and Loop of Henle cells suggests context-specific functional roles in stone pathogenesis. In contrast, the predominant negative correlation (Scissor^−^) observed in fibroblasts implies that their homeostatic, matrix-maintaining functions may be suppressed or altered in the stone-forming microenvironment (Figure 3A,B). These findings indicated that principal cells, Loop of Henle cells, and fibroblasts are key players in CaOx nephrolithiasis. Cell communication analysis of cell subtypes indicated that most of them, including endothelial cells, Loop of Henle cells, and fibroblasts exhibited stronger and more interactions with other cell subtypes in CaOx kidney stones group compared to the control (Figure 3C–F).

### 3.2. ACSL4, PTK2, DUSP4, MMP7, PHLDB2, TGM2, PPT1, SPARCL1, and LTF Were Identified as Key Genes in CaOx Kidney Stones

The intersection of 851 DEGs and 3483 key ERSRGs yielded 538 candidate genes (Figure 4A). Expression analysis highlighted nine key genes, *ACSL4*, *PTK2*, *DUSP4*, *MMP7*, *PHLDB2*, *TGM2*, *PPT1*, *SPARCL1*, and *LTF*, which exhibited a significant difference between CaOx kidney stones and control groups (Figure 4B). Further investigation into their subcellular locations showed that *DUSP4*, *LTF*, *MMP7*, *PHLDB2*, *PPT1*, *SPARCL1*, and *TGM2* were mainly located in cytoplasm, while the mRNA expressions of *ACSL4* and *PTK2* were predominantly in nucleus (Figure 4C, Table 2). It should be clarified that the output results of the mRNALocater reflect the distribution trend of transcripts in cells, rather than the actual localization of corresponding proteins. Correlation analysis revealed that *MMP7* had the strongest positive interaction with *LTF* (cor = 0.74, *p* < 0.05), while *PPT1* significantly negative correlated with *DUSP4* (cor = −0.64, *p* < 0.05) (Figure 4D). The top 20 functionally related genes included ACTN1, KCNH2, and CX3CL1, with corresponding functions related to secretory granule lumen, ossification, and cell–matrix adhesion, among others (Figure 4E).

Enrichment analysis of key genes identified 313 GO terms and 8 KEGG pathways (Figure 4F,G). These GO terms included fatty-acyl-CoA biosynthetic/metabolic process, cell–substrate junction, and serine-type endopeptidase activity, while KEGG pathways contained fatty acid metabolism, efferocytosis, and fatty acid biosynthesis (Appendix A).

### 3.3. Key Genes Accurately Predicted the Risk of Caox Kidney Stones

A nomogram was developed incorporating key gene expression data for predicting the risk of CaOx kidney stones (Figure 5A). The slope of the calibration curve was close to ideal curve (Figure 5B), suggesting reliable predictive performance. The area under ROC curve (AUC) value was 0.774, demonstrating that the nomogram had robust predictive accuracy (Figure 5C). Additionally, DCA revealed that the nomogram had a better clinical net benefit (Figure 5D). These findings collectively suggested that the nomogram effectively predicted the risk of CaOx kidney stones. To further explore the functions of key genes in CaOx kidney stones, GSEA was performed. The analysis revealed that a total of 66, 76, 77, 93, 17, 90, 114, 87, and 74 pathways were, respectively, enriched by *ACSL4*, *PTK2*, *DUSP4*, *MMP7*, *PHLDB2*, *TGM2*, *PPT1*, *SPARCL1*, and *LTF* (Figure 5E). Notably, except PTK2, remaining key genes all enriched olfactory transduction, underscoring its significance in CaOx kidney stones through these key genes.

### 3.4. Regulatory Networks and Potential Therapeutic Drugs of Key Genes

Regulatory mechanisms are important for diseases, so we explored the regulatory relationship of key genes. Notably, 110 miRNAs and 49 lncRNAs were predicted based on key genes and constructed a regulatory network comprising 154 nodes and 380 edges (Figure 6A). Notable interaction pairs in this network included ACSL4-hsa-miR-301a-3p-MIR17HG, TGM2-hsa-miR-519d-3p-SNHG16, MMP7-hsa-miR-379-3p-NORAD, PTK2-hsa-miR-7-5p-AL078639.1, and PHLDB2-hsa-miR-664b-3p-AC067930.1. Additionally, a key gene-TF regulatory network was created with 76 nodes and 103 edges (Figure 6B), highlighting interaction pairs such as CTCF-DUSP4/PPT1/PTK2/TGM2 and MYOG-ACSL4/PHLDB2/PTK2/TGM2.

In the search for novel treatments for CaOx kidney stones, key genes were utilized to predict potential drugs, and a key gene–drug network was established with 87 nodes and 81 edges (Figure 6C). Among these drugs, atorvastatin [27], thiazide diuretics [28], and tamsulosin [29] had been reported to be significant for the treatment of nephrolithiasis. According to the intersection score of key genes with drugs, molecular docking analysis was conducted across ACSL4-tilfrinib, PTK2-VS-4718, MMP7-RS 39066, TGM2-levamisole, PPT1-plitidepsin, and LTF-talactoferrin alfa. The results showed that all binding energies were below −5 kcal/mol (Table 3), highlighting the strong binding affinities. Specifically, tilfrinib interacted with *ACSL4* residue PRO-618, while vs-4718 associated with *PTK2* residue LYS-1017, and rs 39066 correlated with *MMP7* residue PRO-238, GLU-219, ALA-182, HIS-218, TYR-240, ARG-202, ASN-179, and LEU-181. Plitidepsin associated with *PPT1* residue SER-50 and ASN-47, whereas talactoferrin alfa correlated with *LTF* residue GLY-323 and SER-322 (Figure 6D). It is noted that these are computational predictions intended to generate hypotheses for future functional and pre-clinical validation.

### 3.5. The Differential Expression of Key Genes in Key Cell Subtypes

To explore the differentiation of endothelial cells, pseudotime analysis was performed. The results showed that all differentiation states were present in CaOx kidney stones, whereas control samples only exhibited the early differentiation stage (Figure 7A,B). All key genes were expressed in both early and late stages of differentiation (Figure 7C), with *SPARCL1* showing higher expression than other genes. Secondary clustering analysis of endothelial cells identified four key cell subtypes: lymphatic cell, glomerular capillary endothelial cell, afferent/efferent arteriole endothelial cell, and ascending vasa recta endothelial cell (Figure 7D,E). Proportion analysis revealed that glomerular capillary endothelial cells were the most abundant in both control (42.3%) and CaOx kidney stone samples (46.6%) (Figure 7F). Further examination of key gene expression in these subtypes showed significant differences between control and CaOx kidney stone groups for *LTF*, *MMP7*, and *SPARCL1* across all key cell subtypes. *ACSL4* exhibited marked differences in lymphatic cells, *DUSP4* in lymphatic, glomerular capillary, and ascending vasa recta endothelial cells, *PHLDB2* in lymphatic and afferent/efferent arteriole endothelial cells, *PPT1* and *PTK2* in lymphatic cells, and *TGM2* in afferent/efferent arteriole endothelial cells (Figure 7G).

### 3.6. Key Gene mRNA Expression (qRT-PCR)

Compared to the control group, HK-2 cells exposed with CaOx exhibited significantly increased mRNA expression levels of *ACSL4*, *MMP7*, *TGM2*, *PPT1*, and *LTF*. Conversely, mRNA expression levels of *PTK2*, *DUSP4*, *SPARCL1*, and *PHLDB2* were significantly decreased (Figure 8). These qRT-PCR results validated the bioinformatics findings.

### 3.7. Key Protein Expression Analysis (Western Blotting)

Compared to the control group, HK-2 cells exposed with CaOx exhibited significantly increased protein expression levels of *ACSL4*, *MMP7*, *TGM2*, *PPT1*, and *LTF*. Conversely, protein expression levels of *PTK2*, *DUSP4*, *SPARCL1*, and *PHLDB2* were significantly decreased (Figure 9). These Western blotting results are consistent with the findings from the aforementioned qRT-PCR analysis and bioinformatics analysis.

## 4. Discussion

Kidney stones are a highly prevalent urological condition, with CaOx stones being the predominant type, accounting for over 80% of all kidney stone cases [30]. This condition is not only a source of severe pain but can also result in significant kidney damage. Research indicates that oxidative stress, inflammation, and autophagy-related cell death are primary drivers of CaOx crystal deposition, which subsequently exacerbates kidney injury [31]. ERS, is a common form of cellular stress and centrally involved in the development and progression of CaOx kidney stones. Our research group previously demonstrated that ERS is hyperactivated in CaOx kidney stones and that modulating ERS can substantially improve kidney function and reduce crystal deposition [11,12,32,33]. Therefore, targeting ERSRGs represents a promising direction worthy of further exploration for developing novel treatments against CaOx kidney stones. In this study, we initially examined CaOx kidney stone samples and determined that endothelial cells were the key cells with the highest ERSRGs scores. Under normal conditions, endothelial cells primarily serve as a protective barrier [34]. However, in chronic kidney disease, endothelial ERS may initiate stone formation by promoting Endothelial-to-Mesenchymal Transition (EndMT), a key process in the vascular pathology of Randall’s plaques, the subepithelial calcifications that anchor CaOx stones [35,36,37]. And, meanwhile, the significant enrichment of “fluid shear stress and atherosclerosis” pathways among endothelial ERSRGs highlights their role in sensing hemodynamic changes and creating a pro-inflammatory milieu, which subsequently predisposes the tubules to crystal injury and recruits/activates macrophages [38]. Furthermore, the identification relied on the average score from five distinct algorithms. Thus, while tubular epithelial cells are the primary site of crystal interaction, endothelial cells may play a pivotal role in early microenvironmental remodeling. Therapeutic strategies targeting these cells have significant potential in the understanding and regulation of kidney stone disease.

Further analysis identified nine key genes related to ERS in endothelial cells: *ACSL4*, *PTK2*, *DUSP4*, *MMP7*, *PHLDB2*, *TGM2*, *PPT1*, *SPARCL1*, and *LTF*. Notably, *ACSL4* (acyl-CoA synthetase long chain family member 4) is a key initiator of ferroptosis and is primarily found within mitochondria, peroxisomes, and the ER, where it incorporates coenzyme A into arachidonic acid [39,40]. The expression of *ACSL4* is upregulated in CaOx kidney stones [41,42], closely linking it to kidney injury and fibrosis. Our study confirmed this upregulation, further highlighting ACSL4’s significant role in CaOx kidney stones [42]. *MMP7* (matrix metalloproteinase 7) is associated with active stone disease and endomineralization. It breaks down collagen and proteoglycans and serves as a significant indicator of kidney fibrosis [43]. Urinary *MMP7* levels were significantly elevated in patients with stone disease versus healthy controls [44]. It is upregulated in nearly all kidney diseases and serves as a prominent biomarker for kidney-related conditions. *PHLDB2* (pleckstrin homology-like domain family B member 2) contains a PH domain involved in protein complex formation and cell migration [45] and is associated with overall survival in renal cell carcinoma [46]. *TGM2* (transglutaminase 2) is a Ca^2+^-dependent enzyme involved in processes such as wound healing, apoptosis, and inflammation, and is a biomarker for chronic kidney disease [47]. It also plays a role in hyperglycemia-induced diabetic nephropathy [48]. *SPARCL1* (secreted protein acidic and cysteine-rich like 1), a member of the SPARC family, may act as a tumor suppressor in renal cell carcinoma, regulating disease progression through the p38/JNK/ERK pathway [49]. We found that *SPARCL1* expression was significantly reduced in CaOx kidney stones, suggesting its potential impact on disease progression. *PTK2* (protein tyrosine kinase 2) is a non-receptor tyrosine kinase involved in cell adhesion, migration, and proliferation, positively regulating cell population growth, ubiquitin-dependent protein degradation, and protein phosphorylation [50]. Moreover, *DUSP4* (dual specificity phosphatase 4) overexpression in endothelial cells prevents apoptosis induced by hypoxia/reoxygenation through eNOS upregulation [51]. *PPT1* (palmitoyl protein thioesterase-1) is a lysosomal enzyme responsible for the protein depalmitoylation, which is crucial for organelle function, lipid metabolism, and Ca^2+^ transport [52]. *LTF* (Lactotransferrin) is an antimicrobial, anti-inflammatory, and antioxidant factor that provides a natural defense against damaging stimuli [53]. These findings collectively underscore the importance of key genes in kidney-related diseases. Nonetheless, the current research landscape is heavily skewed towards investigating these genes in the context of cancers [54,55]. Our pseudotime analysis was further designed to explore the dynamic expression patterns of these identified hub genes across a continuum of endothelial cell states. The analysis successfully revealed that these key genes are expressed throughout the inferred differentiation trajectory, with their expression levels changing in a coordinated manner. This suggests that the biological processes governed by these genes are active across multiple stages of endothelial cell state transitions in the context of CaOx kidney stones. However, their specific contributions to CaOx kidney stones remain underexplored. Our findings establish a potential association but cannot rule out that these genes are part of a broader injury response. Therefore, further studies are needed to clarify their roles.

Utilizing these key genes, we developed a nomogram to evaluate the risk of CaOx kidney stones. The model demonstrated robust predictive power and accuracy, achieving an AUC value of 0.774. DCA further revealed a substantial net clinical benefit. A comparable nomogram used in a prior study for assessing CaOx kidney stone risk also exhibited an AUC of 0.772 and a high net clinical benefit [56]. These findings collectively suggest that our selected key gene combinations offer enhanced efficacy in risk prediction. Additionally, we conducted functional enrichment analysis on these key genes and observed that, aside from *PTK2*, all of the other genes were enriched in olfactory transduction pathway. While olfactory receptors were historically believed to be predominantly located in the olfactory epithelium [57], emerging evidence indicates their presence in non-sensory organs, including the kidney [58]. Mechanistically, Zhang et al. demonstrated that CCL7 may promote the progression of kidney stone disease by activating the olfactory transduction pathway, thereby facilitating the deposition of CaOx and calcium phosphate (CaP) [59]. Furthermore, the differential expression of several OR genes (OR10A5, OR9A2, and OR1L3) has been linked to variations in glomerular filtration rate (GFR) and renin release, implicating this pathway in key renal physiological and pathophysiological processes [59]. Building on computational simulations and in vivo studies, Ali Motahharynia et al. identified five olfactory receptors (ORs) (including Olfr433) that are significantly associated with the progression of kidney fibrosis, suggesting their potential role in inflammatory responses and myofibroblast generation [59,60]. Hence, we propose the following testable hypothesis based on our data and the published literature: the observed pathway enrichment may reflect the activation of one or more ectopic olfactory receptors in renal tubular or endothelial cells upon CaOx crystal injury, leading to the production of pro-inflammatory cytokines, which in turn exacerbate crystal adhesion, macrophage recruitment, and tubular damage, warranting future functional validation. Collectively, these insights underscore the potential significance of olfactory transduction in kidney-related diseases.

To sum up, our research innovatively combines single-cell and transcriptomic analyses to pinpoint key genes linked to ERS in CaOx kidney stones. These genes offer fresh insights into the disease’s mechanisms and lay a foundation to assess their potential for therapeutic intervention. However, the present study has several limitations that should be considered. First, our findings are primarily derived from bioinformatics analyses. Although we identified key genes and pathways—such as the role of the olfactory receptor signaling pathway in renal CaOx injury and nine potential therapeutic targets—these results remain computationally derived and their precise temporal and causal relationships as well as endothelial ERS with crystal formation remain to be established (e.g., using endothelial-specific ERS models or in vitro co-culture systems). The definitive demonstration of causality, such as through gene knockout or knockdown models in a CaOx context, is essential to determine whether these genes are specifically required for stone pathogenesis or merely part of a generalized injury response. Future studies using spatial transcriptomics or in situ validation are needed to directly assess whether their expression is localized to areas of crystal deposition or specific injury patterns unique to nephrolithiasis. Similarly, while molecular docking provided computational support for drug–gene interactions, these predictions require validation through control docking simulations, in vitro binding assays, and in vivo efficacy studies. It is crucial to clarify that this work is hypothesis-generating, and future functional studies (e.g., gene modulation in disease models) are essential to formally test if targeting these genes alters disease outcomes. Second, the single-cell RNA-seq dataset, though informative, was limited in sample size (*n* = 3 per group), which may constrain the detection of subtle heterogeneity or rare cell states. Furthermore, while pseudotime analysis suggested dynamic endothelial transitions, future lineage-tracing or in vitro differentiation models are needed to formally validate the inferred trajectory. Separate pseudotime analyses on pre-defined endothelial subtypes would also help clarify the relationship between subtype identity and differentiation state. Although the potential role of endothelial cells opens a new perspective for understanding the multifaceted pathogenesis of kidney stones, the current study cannot distinguish between greater ER or metabolic activity versus disease-specific activity in these cells. Third, the predictive model and nomogram were constructed using a public dataset (GSE73680) that lacks detailed clinical variables such as urinary oxalate, calcium, or kidney function. Their integration in future studies is critical to improve clinical relevance and predictive power. Moreover, the nomogram should be regarded as a preliminary auxiliary tool rather than a standalone clinical predictor, pending validation in larger, multi-center cohorts. Finally, our findings are based on Chinese cohorts, and their generalizability across ethnic populations requires further investigation. Although core pathways like ERS and ferroptosis are fundamental biological processes, population-specific differences may modulate their roles. In future expanded cohorts, applying stricter multiple testing corrections—including adjustments for correlation analyses, differential expression, and enrichment *p*-values—will further strengthen the robustness of the identified gene signatures. In summary, this work serves as a foundational, hypothesis-generating resource. We emphasize that our results establish association rather than causation, and we explicitly call for future functional experiments, independent replications, and clinical integrations to validate and extend the proposed mechanisms.

## 5. Conclusions

In this study, we identified endothelial cells as key cellular components in CaOx kidney stones and pinpointed nine key genes associated with ERS: *ACSL4*, *PTK2*, *DUSP4*, *MMP7*, *PHLDB2*, *TGM2*, *PPT1*, *SPARCL1*, and *LTF*. The expression of these genes differed significantly between CaOx and control groups. The nomogram constructed was based on these key genes and demonstrated robust predictive performance for CaOx kidney stone risk. Additionally, functional enrichment analysis showed that, except *PTK2*, the remaining genes were enriched in olfactory transduction pathway, highlighting its potential significance in CaOx kidney stones. Expression analyses with secondary clustering of endothelial cells further revealed significant differences in the expression of *LTF*, *MMP7*, and *SPARCL1* across key cell subtypes in both disease and control groups, underscoring the importance of these genes. Collectively, these findings provide a theoretical foundation for identifying new targets in CaOx kidney stones.

## Figures and Tables

**Figure 1 genes-16-01338-f001:**
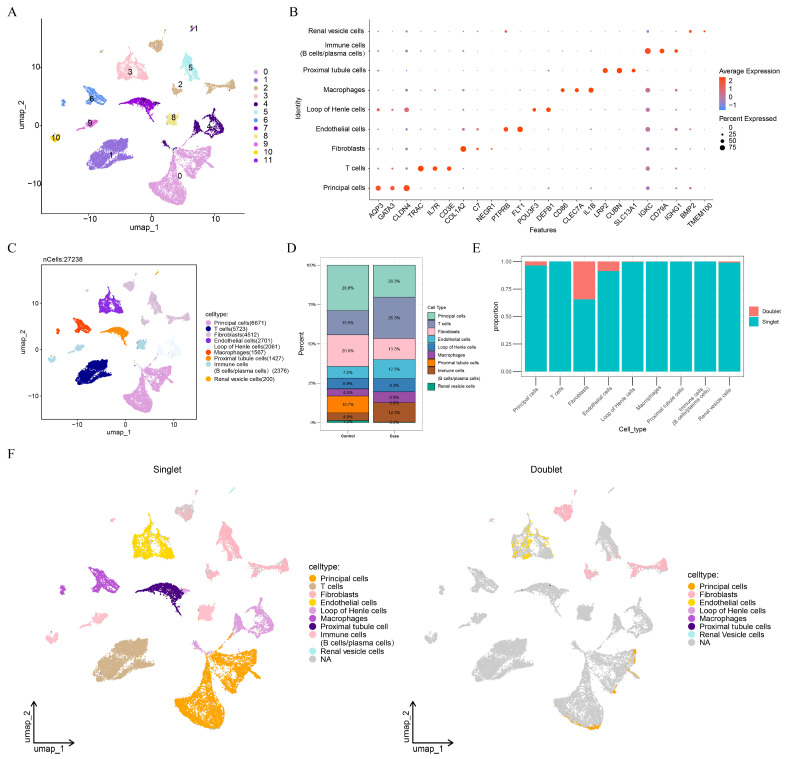
Single-cell analysis for calcium oxalate (CaOx) kidney stones. (**A**) The identification of cell clusters in CaOx kidney stones. (**B**) Marker genes of cell subtypes. (**C**) The annotation of cell subtypes. (**D**) The proportion of cell subtypes in CaOx kidney stones and control samples. (**E**) Assessment of doublet proportions across identified cell subtypes. (**F**) Uniform manifold approximation and projection (UMAP) visualization of the cell subtypes after doublet removal.

**Figure 2 genes-16-01338-f002:**
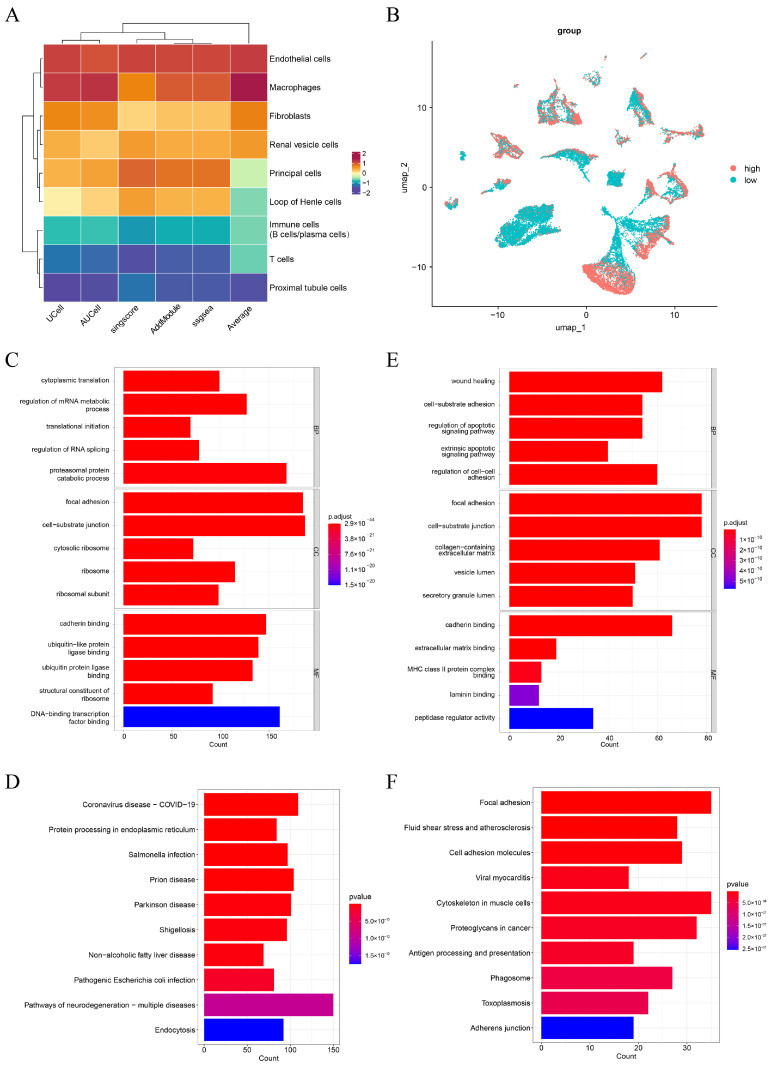
Endothelial cells were selected as key cells in CaOx kidney stones. (**A**) Endoplasmic reticulum stress-related genes (ERSRGs) scores of cell subtypes by five algorithms. (**B**) The distribution of high- and low-score groups of key cells. (**C**,**D**) Gene ontology (GO) and Kyoto Encyclopedia of Genes and Genomes (KEGG) enrichment analyses of key ERSRGs are shown in panels (**C**,**D**), respectively. (**E**,**F**) GO and KEGG enrichment analyses of differentially expressed genes (DEGs) between the high-scoring and low-scoring groups are presented in panels (**E**,**F**), respectively.

**Figure 3 genes-16-01338-f003:**
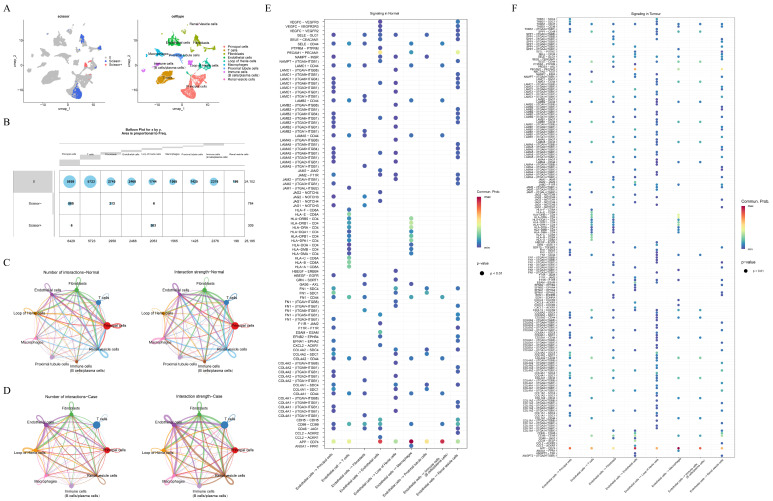
Scissor analysis and cell communication. (**A**) The Scissor clustering diagram of all cell subtypes. (**B**) The cell numbers of different Scissor groups. (**C**,**D**) Cell communication of all cell subtypes in control (**C**) and CaOx kidney stones groups (**D**). (**E**,**F**) The communication of endothelial cells with other cell subtypes in control (**E**) and CaOx kidney stones groups (**F**).

**Figure 4 genes-16-01338-f004:**
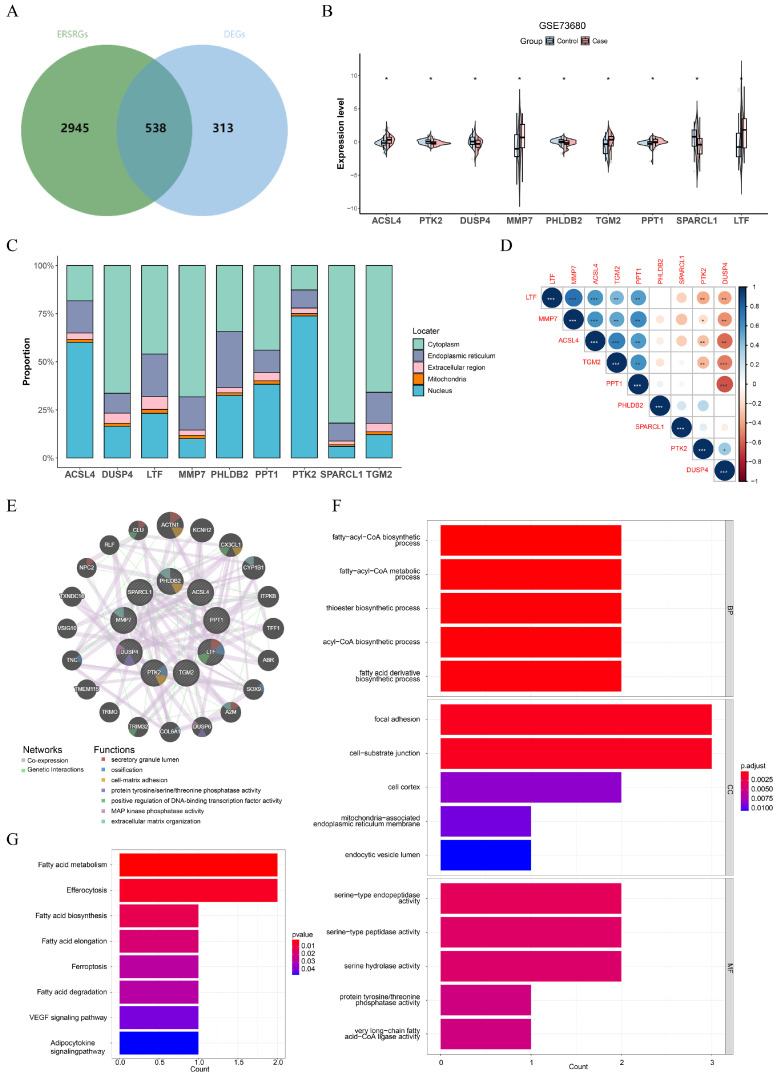
Identification and functional characterization of nine key genes based on their differential expression patterns between CaOx kidney stones and control groups. (**A**) Venn diagram illustrating the intersection of key ERSRGs and DEGs. (**B**) Differential expression of key genes between control and CaOx kidney stones groups in the GSE73680 dataset (Wilcoxon test: *p* < 0.05). (**C**) Predicted mRNA subcellular localization of the key genes using the mRNALocater database. (**D**) Correlation analysis among key genes. (**E**) The prediction of genes functionally correlated with key genes. (**F**,**G**) GO and KEGG enrichment analyses of key genes are presented in panels (**E**,**F**), respectively. (* *p* < 0.05, ** *p* < 0.01, *** *p* < 0.001).

**Figure 5 genes-16-01338-f005:**
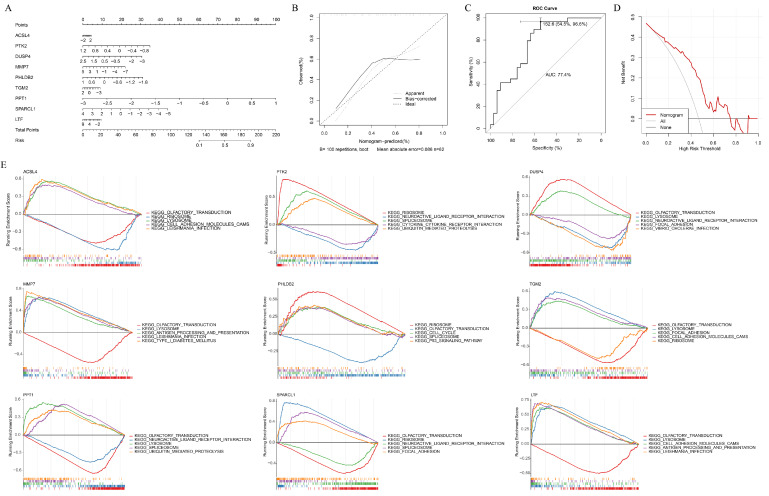
Construction of nomogram and gene set enrichment analysis (GSEA). (**A**) Nomogram of key genes. (**B**–**D**) Calibration curve (**B**), receiver operating characteristic (ROC) (**C**), and decision curve analysis (DCA) (**D**) for nomogram. (**E**) GSEA results of key genes.

**Figure 6 genes-16-01338-f006:**
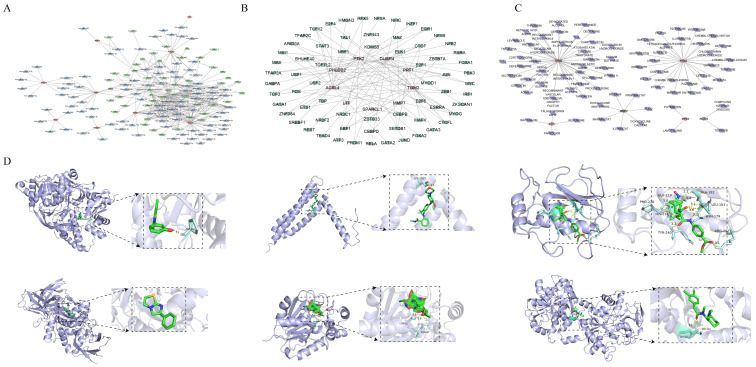
Regulatory networks and potential therapeutic drugs of key genes. (**A**) Key gene-microRNA (miRNA)-long noncoding RNA (lncRNA) regulatory network. (**B**) Key gene (red)-transcription factors (TFs, green) regulatory network. (**C**) Key gene–drug network. (**D**) Molecular docking analysis of key genes with drugs. The protein structure is shown in blue, the small molecule ligand in green, key amino acid residues in light green, and important interacting atoms (hydrogen bonds) highlighted in red.

**Figure 7 genes-16-01338-f007:**
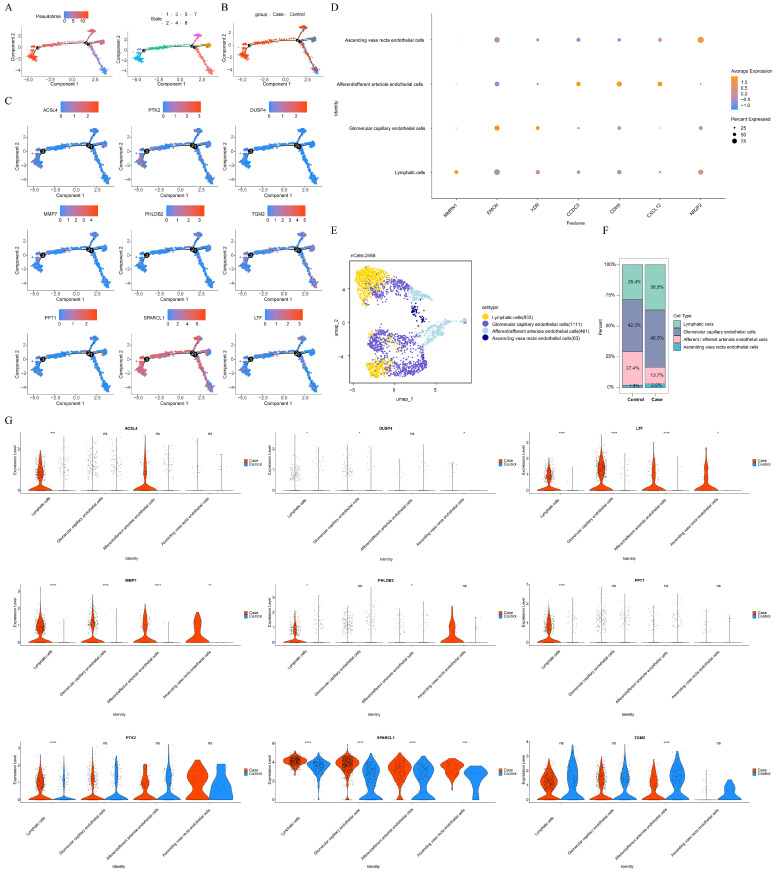
The differential expression of key genes in key cell subtype. (**A**) Pseudotime analysis of endothelial cells. (**B**) The distribution of endothelial cells differentiation in control and CaOx kidney stones. (**C**) The expression of key genes in endothelial cells differentiation. For A-C, numbers with black circles denote critical branching events along the pseudotime axis, indicating a bifurcation in cell fate decisions. (**D**) Marker genes of endothelial cell subtypes. (**E**) The annotation of endothelial cell subtypes. (**F**) The proportion of endothelial cell subtypes. (**G**) The expression of key genes in endothelial cell subtypes. (ns: no significance, * *p* < 0.05, ** *p* < 0.01, *** *p* < 0.001, **** *p* < 0.0001).

**Figure 8 genes-16-01338-f008:**
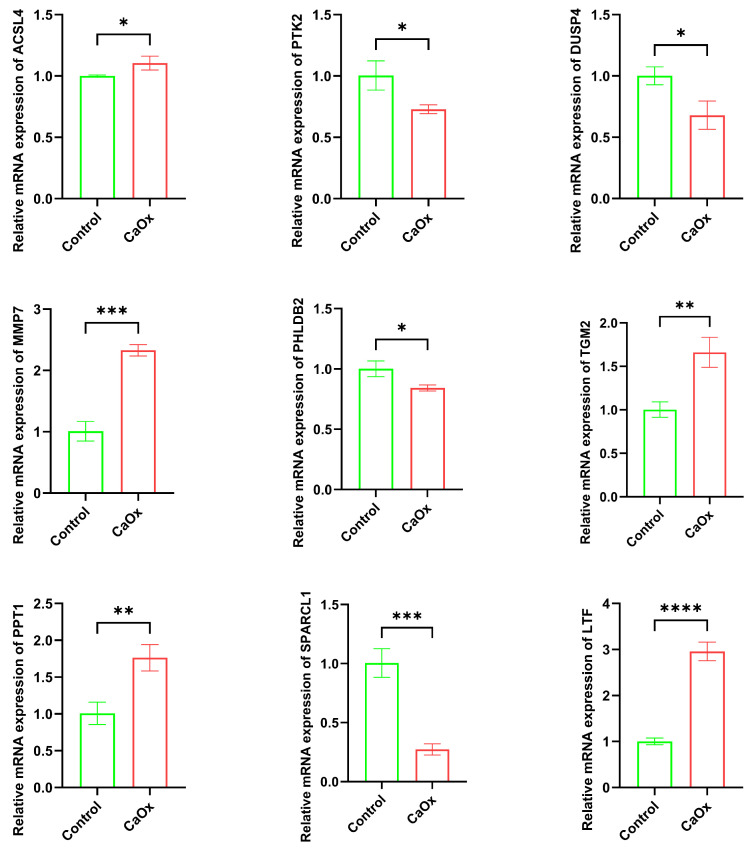
Expression levels of key gene mRNAs. The results of qRT-PCR analysis of key genes. (* *p* < 0.05, ** *p* < 0.01, *** *p* < 0.001, **** *p* < 0.0001).

**Figure 9 genes-16-01338-f009:**
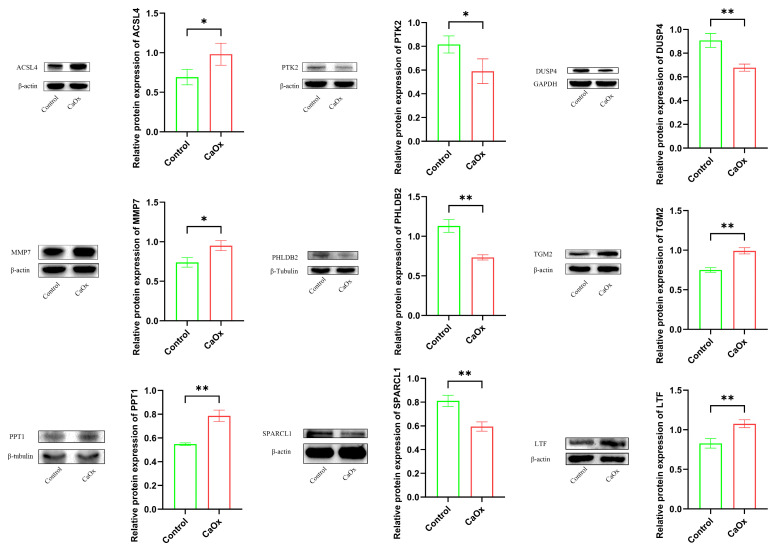
The protein expression of key genes. The results of Western blotting analysis of key genes. (* *p* < 0.05, ** *p* < 0.01).

**Table 1 genes-16-01338-t001:** Primer sequences in RT-qPCR.

Gene Name	Forward Sequence (5′ → 3′)	Reverse Sequence (5′ → 3′)
*ACSL4*	GTGAAAGAATACCTGGACTGG	AGAGAGTGTAAGCGGAGAAG
*PTK2*	CAGTATTGACAGGGAGGATG	AGGCGGTTTCTTTGGTGGAG
*DUSP4*	GCGTCAGTCCAATAGGTCAG	CAGAAACTTCCCATCACCAG
*MMP7*	AGCTCATGGGGACTCCTACC	GTCCAGCGTTCATCCTCATC
*PHLDB2*	CTGAATATCAACGGAACATCG	TCTCTCTGAGCCTGCTGAAC
*TGM2*	GAAGGAGGAGACAGGGATGG	CAGCGGTGTTGTTGGTGATG
*PPT1*	GTATCGCAACCACAGCATCT	TCCGAATCTACAGGGTCCAC
*SPARCL1*	ACGGTAGCACCTGACAACAC	ATGGTGGGAATCGTCTTCTGT
*LTF*	GAACCGTACTTCAGCTACTCTG	CTCATACTCGTCCCTTTCAGC
*β-actin*	CCTGGCACCCAGCACAAT	GGGCCGGACTCGTCATAC

**Table 2 genes-16-01338-t002:** The mRNA-subcellular location of key genes utilizing mRNALocater database (supplemented by Uniprot annotation).

ID	FastaID	mRNALocater	Cytoplasm	Endoplasmic Reticulum	Extracellular Region	Mitochondria	Nucleus	Protein Localization Annotation (From Uniprot)
ACSL4	NC_000023.11:c109733257-109641335	Nucleus	0.1836	0.167	0.0336	0.0152	0.6006	Mitochondrion outer membrane, Microsome membrane, Endoplasmic reticulum membrane, Cell membrane
DUSP4	NC_000008.11:c29350684-29333064	Cytoplasm	0.6639	0.1029	0.0533	0.0151	0.1648	Nucleus
LTF	NC_000003.12:c46485234-46435645	Cytoplasm	0.46	0.2209	0.0668	0.0204	0.2318	Isoform 1, Secreted, Cytoplasmic granule, Note: Secreted into most exocrine fluids by various endothelial cells. Stored in the secondary gran-ules of neutro-phils. Isoform DeltaLf, Cyto-plasm, Nucleus, Note: Mainly lo-calized in the cy-toplasm.
MMP7	NC_000011.10:c102530747-102520508	Cytoplasm	0.6825	0.1725	0.0277	0.0164	0.1009	Secreted, extracel-lular space, extra-cellular matrix
PHLDB2	NC_000003.12:111732496-111976517	Cytoplasm	0.3434	0.2906	0.0277	0.0134	0.3249	Cytoplasm, Cytoplasm, cell cortex, Membrane; Peripheral membrane protein
PPT1	NC_000001.11:c40097252-40071461	Cytoplasm	0.4397	0.1168	0.0428	0.0177	0.383	Cell projection, podosome, Note: Translocates to the plasma membrane at high levels of PtdIns-(3,4,5)-P3. At low levels of PtdIns-(3,4,5)-P3 is translocated to vesicular compartments. Localized to the myotube podosome cortex that surrounds the core Lysosome, Secreted, Golgi apparatus, Endoplasmic retic-ulum
PTK2	NC_000008.11:c141002079-140657900	Nucleus	0.1269	0.0943	0.0287	0.0123	0.7378	Cell junction, focal adhesion, Cell membrane by simi-larity; Peripheral membrane protein, Cytoplasm, peri-nuclear region, Cy-toplasm, cell cortex, Cytoplasm, cyto-skeleton, Cyto-plasm, cytoskeleton, microtubule organ-izing center, cen-trosome, Nucleus, Cytoplasm, cyto-skeleton, cilium ba-sal body, Cyto-plasm, Note: Con-stituent of focal ad-hesions. Detected at microtubules.
SPARCL1	NC_000004.12:c87529376-87473335	Cytoplasm	0.8185	0.0933	0.0195	0.0092	0.0595	Secreted, extracellular space, extracellular matrix
TGM2	NC_000020.11:c38168475-38127385	Cytoplasm	0.6584	0.1614	0.0438	0.0153	0.121	Cytoplasm, cytosol, Nucleus, Chromosome, Secreted, extracellular space, extracellular matrix, Cell membrane, Mitochondrion, Note: Mainly localizes to the cytosol, present at much lower level in the nucleus and chromatin, Also secreted via a non-classical secretion pathway to the extracellular matrix, Isoform 2, Cytoplasm, perinuclear region

Uniprot Annotation information was soured from https://www.uniprot.org/, (accessed on 28 October 2025).

**Table 3 genes-16-01338-t003:** Molecular docking of key genes with drugs.

Gene Symbol	UniProt	Chemical Name	Kcal/mol
*ACSL4*	O60488	TILFRINIB	−8.3
*PTK2*	Q05397	VS-4718	−7
*MMP7*	P09237	RS 39066	−7.9
*TGM2*	P21980	LEVAMISOLE	−5.9
*PPT1*	P50897	PLITIDEPSIN	−12.8
*LTF*	P02788	TALACTOFERRIN ALFA	−7.5

## Data Availability

Data are available from the corresponding author upon reasonable request.

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
