# Peer review of "Identification of Key Genes Associated with Endoplasmic Reticulum Stress in Calcium Oxalate Kidney Stones"

_genes, 2025, doi:10.3390/genes16111338_

Round 1

Reviewer 1 Report

Comments and Suggestions for Authors

General assessment:

The manuscript presents a novel and well-structured bioinformatics and experimental study investigating endoplasmic reticulum stress (ERS)-related genes in calcium oxalate kidney stones. The integration of scRNA-seq and transcriptome datasets with downstream experimental validation is appropriate, and the visualizations are generally clear and informative. The use of publicly available GEO data is commendable, and the analytical flow is logical and easy to follow. However, several methodological and presentation issues should be addressed to ensure reproducibility and improve clarity before publication.

Major comments

  1. Missing list of ER-stress-related genes (ERSRGs)
    • The manuscript states that 551 ERS-related genes were obtained from GeneCards (relevance score ≥ 10), but the full gene list is not provided as supplementary material.
    • Please include a supplementary table listing all 551 genes with their identifiers (gene symbols, Entrez/Ensembl IDs) and specify the GeneCards query date and exact criteria used. This is essential for reproducibility.
  2. CellPhoneDB version clarification
    • The Methods section lists CellPhoneDB v1.6.1. The latest widely used release is v4–5. Please clarify why an older version was used and describe any technical considerations or compatibility reasons that justified this choice. Providing the specific release link or commit hash would improve reproducibility.
  3. Subcellular localization inconsistency and methodological clarification
    • Table 2 indicates that ACSL4 and PTK2 are predominantly nuclear according to mRNALocater predictions. However, the Discussion section correctly describes ACSL4 as primarily localized in the endoplasmic reticulum, mitochondria, and peroxisomes, consistent with existing literature (e.g., Biology (Basel). 2023 Jun 15;12(6):864. doi: 10.3390/biology12060864).
    • Please reconcile this discrepancy and clarify that mRNALocater predicts mRNA rather than protein localization. Protein localization should instead rely on experimental or database evidence (e.g., UniProt, HPA). Table 2 should be corrected accordingly.

Minor comments

4. Typographical correction

  • Please correct the typo “Monole2” to Monocle2 in the Methods section.

5. Figure 2 legend labeling

  • The legend currently lists “GO (C) and KEGG (C)”. Please revise the wording and indications of legend for ensure clarity.

6. Figure 3E resolution

  • The image quality of Figure 3E is low, making it difficult to assess the intercellular communication patterns. Please provide a higher-resolution version or vector-based figure to allow proper evaluation.

7. General comments on figures and presentation

  • Ensure consistent capitalization and labeling across figures and legends (e.g., “Principal cells”, “Endothelial cells”).

Summary

This manuscript provides a valuable contribution by identifying potential ERS-related target genes in calcium oxalate nephrolithiasis using an integrative multi-omics approach. The findings are potentially relevant for understanding kidney stone pathogenesis. Addressing the issues above particularly the missing ERSRG list, the subcellular localization clarification, and figure labeling will substantially improve the manuscript’s rigor and clarity.

Author Response

1. Summary

Dear Reviewer,

Thank you for your thoughtful suggestions and insights, which have benefited from the manuscript. I am looking forward to working with you to move this manuscript closer to publication in Genes.

The manuscript has been rechecked and the necessary changes have been made in accordance with your suggestions. The modified sections are highlighted in red within the manuscript. The responses to all comments have been prepared and attached below. We have tried our best to solve the problems you proposed, and we hope that the revised manuscript is now suitable for publication in the Genes. If you have any questions remained about this paper, please feel free to contact us.

2. Point-by-point response to Comments and Suggestions for Authors

Comments 1: [General assessment:

The manuscript presents a novel and well-structured bioinformatics and experimental study investigating endoplasmic reticulum stress (ERS)-related genes in calcium oxalate kidney stones. The integration of scRNA-seq and transcriptome datasets with downstream experimental validation is appropriate, and the visualizations are generally clear and informative. The use of publicly available GEO data is commendable, and the analytical flow is logical and easy to follow. However, several methodological and presentation issues should be addressed to ensure reproducibility and improve clarity before publication.

Major comments

1.     Missing list of ER-stress-related genes (ERSRGs)

The manuscript states that 551 ERS-related genes were obtained from GeneCards (relevance score ≥ 10), but the full gene list is not provided as supplementary material.

Please include a supplementary table listing all 551 genes with their identifiers (gene symbols, Entrez/Ensembl IDs) and specify the GeneCards query date and exact criteria used. This is essential for reproducibility.]

Response 1: We appreciate the reviewer's comment regarding reproducibility. In response, we have created Supplementary Table 1, which provides the list of the 551 genes (with gene symbols, Uniprot ID, GC ID, Entrez/Ensembl IDs) sourced from GeneCards. The query was performed on June 18, 2025, and the sole filtering criterion was a Relevance score ≥ 10. The revised manuscript has been updated based on your feedback, with detailed modifications outlined on line 96-97.

Comments 2: [2. CellPhoneDB version clarification

The Methods section lists CellPhoneDB v1.6.1. The latest widely used release is v4–5. Please clarify why an older version was used and describe any technical considerations or compatibility reasons that justified this choice. Providing the specific release link or commit hash would improve reproducibility.]

Response 2: Thank you for raising this important point regarding the version of CellPhoneDB. We sincerely appreciate the need for methodological transparency and reproducibility.

The primary reason for using CellPhoneDB v1.6.1 was to ensure direct comparability and consistency with the established analytical pipeline and reference data from the original CellPhoneDB methodology at the time our study was initiated. Our initial analyses were built upon this specific version to guarantee the reliability and interpretability of our cell-cell interaction results. While newer versions (v4–v5) of CellPhoneDB are available and include valuable updates, the core functionality employed in our analysis—statistical assessment of ligand–receptor interactions from single-cell RNA sequencing data—remains robust and valid in v1.6.1. The fundamental principles are consistent across versions. To further support reproducibility, we have provided the link to CellPhoneDB v1.6.1 used in this study: https://github.com/sqjin/CellChat. The revised manuscript has been updated based on your feedback, with detailed modifications outlined on line 136.

Comments 3: [3. Subcellular localization inconsistency and methodological clarification

Table 2 indicates that ACSL4 and PTK2 are predominantly nuclear according to mRNALocater predictions. However, the Discussion section correctly describes ACSL4 as primarily localized in the endoplasmic reticulum, mitochondria, and peroxisomes, consistent with existing literature (e.g., Biology (Basel). 2023 Jun 15;12(6):864. doi: 10.3390/biology12060864).

Please reconcile this discrepancy and clarify that mRNALocater predicts mRNA rather than protein localization. Protein localization should instead rely on experimental or database evidence (e.g., UniProt, HPA). Table 2 should be corrected accordingly.]

Response 3: We sincerely thank the reviewer for this critical observation and the opportunity to clarify this important methodological point. The reviewer is correct to highlight the apparent discrepancy between our mRNA localization predictions and the established protein localization for ACSL4 from the literature.

In response to this comment, we have taken the following comprehensive actions to reconcile this difference and enhance methodological transparency:

Methodological clarification: We have explicitly clarified in the Methods section: mRNALocater was utilized to predict mRNA (not protein) localization, given its high-throughput efficiency and reliability in eukaryotes. The resulting predictions—spanning the cytoplasm, endoplasmic reticulum, extracellular region, mitochondria, and nucleus—facilitated subsequent functional annotation and analysis of post-transcriptional regulatory mechanisms (PMID: 33823302) (line 145-150). This upfront statement prevents potential misinterpretation.

Result interpretation correction: In the Results section, we have carefully rephrased our description to specify that our data reflect the "mRNA expressions of ACSL4 and PTK2 were predominantly in nucleus (Figure 4C, Table 2)." We have added a sentence immediately following this to emphasize that "It should be clarified that the output results of mRNALocater reflect the distribution trend of transcripts in cells, rather than the actual localization of corresponding proteins." (line 317-320).

Table 2 enhancement for reproducibility: To directly address the discrepancy and provide complete knowledge for the reader, we have updated Table 2 to include the "Protein localization annotation (from UniProt)" distinct column. This allows for a direct comparison and underscores the fundamental difference between transcript and protein localization. A footnote has been added to the table citing the UniProt source (line 341-342).

We believe these revisions have thoroughly addressed the concern by accurately distinguishing between mRNA and protein localization data, thereby improving the clarity, accuracy, and reproducibility of our manuscript.

Comments 4: [Minor comments

4. Typographical correction

Please correct the typo “Monole2” to Monocle2 in the Methods section.]

Response 4: We thank the reviewer for pointing this out. The typo has been corrected to "Monocle2" in the Methods section (line 187).

Comments 5: [5. Figure 2 legend labeling

The legend currently lists “GO (C) and KEGG (C)”. Please revise the wording and indications of legend for ensure clarity.]

Response 5: We thank the reviewer for this suggestion. The legend for Figure 2 has been revised as recommended to ensure clarity in labeling: (C-D) Gene ontology (GO) and Kyoto Encyclopedia of Genes and Genomes (KEGG) enrichment analyses of key ERSRGs are shown in panels (C) and (D), respectively. (E-F) GO and KEGG enrichment analyses of differentially expressed genes (DEGs) between the high- and low-score groups are presented in panels (E) and (F), respectively (line 289-292).

Comments 6: [Figure 3E resolution

The image quality of Figure 3E is low, making it difficult to assess the intercellular communication patterns. Please provide a higher-resolution version or vector-based figure to allow proper evaluation.]

Response 6: We thank Thank you for pointing out the image quality issue. We have now provided a new version of Figure 3E at a resolution of 600 DPI, which significantly improves the clarity and allows for a proper assessment of the intercellular communication patterns (line 304-305).

Comments 7: [General comments on figures and presentation

Ensure consistent capitalization and labeling across figures and legends (e.g., “Principal cells”, “Endothelial cells”).]

Response 7: We thank the reviewer for this comment. We have thoroughly checked all figures and their corresponding legends to ensure consistent capitalization and labeling (e.g., "Principal cells", "Endothelial cells", "Loop of Henle cells") throughout the manuscript.

Comments 8: [Summary

This manuscript provides a valuable contribution by identifying potential ERS-related target genes in calcium oxalate nephrolithiasis using an integrative multi-omics approach. The findings are potentially relevant for understanding kidney stone pathogenesis. Addressing the issues above particularly the missing ERSRG list, the subcellular localization clarification, and figure labeling will substantially improve the manuscript’s rigor and clarity.]

Response 8: We thank the reviewer for their positive assessment of our manuscript as a valuable contribution and for acknowledging the potential relevance of our findings. In our point-by-point response, we have thoroughly addressed all the raised issues, including:

Providing the complete list of ERSRGs (Supplementary Table 1).

Clarifying the distinction between mRNA and protein subcellular localization.

Improving the resolution and labeling of all figures.

We believe these revisions have significantly enhanced the rigor and clarity of the manuscript, and we thank the reviewer again for their constructive feedback.

Reviewer 2 Report

Comments and Suggestions for Authors

Authors provide a broad spectrum analysis of genes involved in kidney stones formation, mainly through bioinformatic methods, and on HK-2 cells, highlighting the role of tubular cells in kidney stones formation. Despite interesting results of this study, some issue can be taken under consideration:

  1. L11-13: please try to improve the sentence and better connect these data,
  2. L13: in the abstract please add that the methodology involved cell culture (HK-2 cells) experiments,
  3. L30 and later: I suggest to not call 'a treatment' an exposition to CaOx og HK-2 cells,
  4. L32-33: please improve the sentence, its a bit too brief,
  5. L82: did you analyze data from 'kidney stones' or 'tissues with stones'? how come cells /proteins were extracted from stones?
  6. L175: please be more specific, did you extract a RNA from HK-2 cells?
  7. L187: similarly, is it related with HK-2 cells experiments?
  8. L217: how different are 'immune cells' from T cells or macrophages mentioned earlier? please be more specific,
  9. L217: please use capital name when talking about the 'loop of Henle',
  10. most Figures are hard to read, please especially try to improve the font,
  11. Figure 4: please improve the title and add more information (when/how were identified)?
  12. L318: drugs related with kidney diseases or nephrolithiasis?
Comments on the Quality of English Language
  1. please re-read the manuscript, some sentences can be improved, e.g. L39-40,
  2. some sentences are written in a way too general way. e.g. howe cells are correlated with kidney stones? (L252-253).

Author Response

1. Summary

Dear Reviewer,

Thank you for your thoughtful suggestions and insights, which have benefited from the manuscript. I am looking forward to working with you to move this manuscript closer to publication in Genes.

The manuscript has been rechecked and the necessary changes have been made in accordance with your suggestions. The modified sections are highlighted in red within the manuscript. The responses to all comments have been prepared and attached below. We have tried our best to solve the problems you proposed, and we hope that the revised manuscript is now suitable for publication in the Genes. If you have any questions remained about this paper, please feel free to contact us.

2. Point-by-point response to Comments and Suggestions for Authors

Comments 1: [L11-13: please try to improve the sentence and better connect these data,]

Response 1: Thank you for the comment from the reviewer. We have carefully reviewed the sentences in lines 11-13 of the manuscript and revised them according to your suggestion. Please refer to the revised manuscript for detailed changes (lines 11-13).

Comments 2: [L13: in the abstract please add that the methodology involved cell culture (HK-2 cells) experiments,]

Response 2: Thank you for your valuable comments. We have added a description of the cell culture (HK-2 cells) experiments in the abstract to better reflect the experimental methods (lines 23-25).

Comments 3: [L30 and later: I suggest to not call 'a treatment' an exposition to CaOx og HK-2 cells,]

Response 3: Thank you for the reviewer's comment. We have changed "CaOx treatment" to "CaOx exposure" in the manuscript to more accurately describe the experimental condition.

Comments 4: [L32-33: please improve the sentence, its a bit too brief,]

Response 4: Thank you for the reviewer's valuable comment. We have expanded the conclusion section to elaborate on the findings of this study and their significance (lines 35-39).

Comments 5: [L82: did you analyze data from 'kidney stones' or 'tissues with stones'? how come cells /proteins were extracted from stones?]

Response 5: Thank you for the reviewer's concern regarding the sample source. Our analysis utilized the GSE73680 dataset, which included kidney tissue samples from patients with CaOx kidney stones and control samples. As detailed in PMID: 27297950, the researchers obtained these samples by cutting renal papillary mucosal tissue (for nephrectomy) or by obtaining renal papillary tissue samples via biopsy forceps (for PCNL and RIRS). Therefore, the data we analyzed was actually from kidney tissue, not directly from the stones themselves. We have clarified this point in the manuscript. Total RNA was extracted from the tissues in RNAlater using an RNeasy Micro Kit (Qiagen). (lines 92-95).

Comments 6: [L175: please be more specific, did you extract a RNA from HK-2 cells?]

Response 6: Thank you for your valuable feedback and for highlighting the need for greater specificity regarding the source of RNA extraction. We confirm that the total RNA was indeed extracted from HK-2 cells in this study. As described in the Methods section, the HK-2 cells were cultured and then harvested for RNA extraction using the RNA Isolator Total RNA Extraction Reagent (R401-01, Vazyme, China). (lines 201-202).

Comments 7: [L187: similarly, is it related with HK-2 cells experiments?]

Response 7: Thank you for the comment. As the reviewer correctly noted, the cells in this experiment were indeed the HK-2 cells. We have clarified this in the revised manuscript (lines 214-215).

Comments 8: [L217: how different are 'immune cells' from T cells or macrophages mentioned earlier? please be more specific,]

Response 8: We thank the reviewer for this comment. The cluster initially labeled as "immune cells" expresses definitive B cells/plasma cells markers (IGKC, CD79A, IGHG1), which are genes involved in immunoglobulin production, representing the adaptive humoral immune component within the renal tissue. We have revised the manuscript accordingly, replacing the general term "immune cells" with the specific identity "immune cells (B cells/plasma cells)" in the Results section (lines 249) and Figures 1-3.

Comments 9: [L217: please use capital name when talking about the 'loop of Henle',]

Response 9: We appreciate the reviewer's attention to detail. We have carefully reviewed and corrected the capitalization of 'Loop of Henle' throughout the manuscript.

Comments 10: [most Figures are hard to read, please especially try to improve the font,]

Response 10: We thank the reviewer for this comment. We have now improved the readability of all figures by increasing the resolution to 600 DPI as per journal requirements and by enlarging the font sizes used in labels, legends, and annotations to ensure they are clear and easy to read.

Comments 11: [Figure 4: please improve the title and add more information (when/how were identified)?]

Response 11: We thank the reviewer for this suggestion. We have revised the title of Figure 4 to more clearly state how the key genes were identified, and to better reflect the comprehensive functional analyses performed. The new title now reads: Identification and functional characterization of 9 key genes based on their differential expression patterns between CaOx kidney stones and control groups (lines 332-339). We believe this significantly improves the clarity and informativeness of the figure.

Comments 12: [L318: drugs related with kidney diseases or nephrolithiasis?]

Response 12: Thank you for pointing out the error. We have corrected the text to specify that the reported drugs are significant for the treatment of nephrolithiasis, not kidney diseases. The manuscript has been updated accordingly (lines 370-373).

Comments 13: [please re-read the manuscript, some sentences can be improved, e.g. L39-40,]

Response 13: Thank you for the suggestion. We have thoroughly re-read the manuscript, paying particular attention to sentence structure, clarity, and flow throughout the entire document. We have made revisions to improve the overall readability of the manuscript. The introduction, as an example (lines 43-47)"

Comments 14: [some sentences are written in a way too general way. e.g. howe cells are correlated with kidney stones? (L252-253).]

Response 14: We thank the reviewer for this constructive feedback. We have revised the sentences at lines 252-253 to provide a more specific and mechanistic interpretation of the Scissor results, moving beyond a general statement of correlation:

Further analysis using the Scissor algorithm, which links single-cell phenotypes to bulk transcriptomic profiles of the disease, revealed distinct cell-type-specific associations with CaOx kidney stones. The identification of both Scissor+ and Scissor- cells within principal cells and Loop of Henle cells suggests context-specific functional roles in stone pathogenesis. In contrast, the predominant negative correlation (Scissor-) observed in fibroblasts implies that their homeostatic, matrix-maintaining functions may be suppressed or altered in the stone-forming microenvironment (Figure 3A-B). These findings indicated that principal cells, Loop of Henle cells and fibroblasts as key players in CaOx nephrolithiasis (lines 293-301).

We once again extend our sincere gratitude for dedicating your valuable time to reviewing our manuscript and for your insightful suggestions. We have revised and improved the manuscript according to your comments, and your feedback has been instrumental in significantly enhancing its quality. We are very pleased to have incorporated your recommendations and appreciate your recognition of our work, which inspires us to continue our in-depth research. We look forward to your further guidance.

Reviewer 3 Report

Comments and Suggestions for Authors

The manuscript asserts a link between endoplasmic reticulum stress (ERS) and calcium oxalate (CaOx) kidney stones, but ERS is a ubiquitous response across many pathologies. How do the authors justify ERS as a specific mechanistic driver rather than a general stress response secondary to crystal-induced injury?

The identification of endothelial cells as “key cells” is intriguing but somewhat counterintuitive given that CaOx crystals primarily interact with tubular epithelial cells. Can the authors clarify the biological reasoning for this conclusion and rule out computational bias?

The idea that ERS-related gene expression in endothelial cells contributes to stone formation is quite a stretch. Is there any evidence that endothelial dysfunction has a causal effect on CaOx stone formation specifically? 

The Introduction refers to the authors' previous work but does not convey how this paper moves beyond it. What is truly novel about this integration of single-cell and bulk transcriptomics? 

The authors combine scRNA-seq (GSE176155) and bulk RNA-seq (GSE73680). Were these datasets from similar patients, tissues, and processing? If not, how were batch effects mitigated?

Only three disease and three control samples were included in the scRNA-seq dataset. How do the authors justify the statistical robustness of identifying 27,000+ cells and performing extensive differential expression analyses on such a small sample base?

The “Scissor” algorithm integrates single-cell and bulk data, but the rationale for parameter selection and validation of its results are not provided. How were false positives avoided?

Multiple algorithms (AUCell, UCell, ssGSEA, singscore, AddModuleScore) were averaged to determine ERSRG scores. How did the authors confirm that these distinct scoring systems produce consistent and non-redundant results?

The thresholds for DEG selection (|log2FC| > 0.5, p < 0.05) appear arbitrary. Were these cutoffs justified or tested for sensitivity?

How were the 551 ERSRGs from GeneCards with relevance score ≥10 curated? Were any redundant or unrelated stress-related genes manually excluded?

The authors conclude that endothelial cells are “key cells” based on the highest ERSRG score. Could this simply reflect greater ER or metabolic activity in endothelial cells rather than disease relevance?

In the nomogram construction, the model achieved an AUC of 0.774, but no internal or external validation was conducted. How reliable is the predictive model without cross-validation or bootstrapping?

The use of calibration and DCA plots on the same dataset used for model building risks overfitting. Did the authors perform any form of data partitioning?

The enrichment of “olfactory transduction” pathways across most genes seems biologically implausible for kidney stones. Could this reflect annotation bias or pathway redundancy in KEGG analysis?

The pseudotime analysis suggests different endothelial differentiation states, but only inferred from static transcriptomic data. Were these differentiation states biologically verified or supported by marker gene expression?

Many of the identified key genes (e.g., ACSL4, MMP7, TGM2) are known mediators of oxidative stress and fibrosis in multiple organ systems. How do the authors distinguish kidney stone-specific roles from general stress or injury responses?

The claim that olfactory receptor pathways influence kidney stone formation lacks mechanistic grounding. Can the authors propose a testable hypothesis or cite direct evidence of olfactory receptor signaling in renal CaOx injury?

PTK2 and SPARCL1 are implicated in cell adhesion and migration. Could their expression changes simply reflect tissue remodeling rather than disease-specific regulation?

The discussion attributes several functional roles to each gene (e.g., ferroptosis, apoptosis, inflammation), but no causal experiments are presented. Are these interpretations speculative or supported by pathway-level validation?

The observed upregulation and downregulation patterns in qRT-PCR and Western blot were consistent, but only in HK-2 cells exposed to CaOx. How do the authors justify using tubular cells to validate genes identified primarily in endothelial cells?

The validation experiments are limited to mRNA and protein expression in a single cell line (HK-2). Why were primary endothelial cells not used, given that they were identified as the “key cell type”?

Were all antibodies and primers validated for specificity and efficiency, particularly for closely related gene families such as MMP7 or ACSL4?

The sample size for in vitro validation (n=3 biological replicates) is minimal. Were these experiments repeated independently to confirm reproducibility?

No in vivo validation (e.g., animal model) was attempted to assess gene function. Without functional perturbation (knockdown/overexpression), can the identified genes truly be termed “key”?

For molecular docking, how were docking poses and binding sites validated? Were any control docking simulations (random or non-target proteins) conducted to estimate false-positive rates?

Given the large number of DEGs and ERSRGs intersected (538 candidates), was multiple testing correction (e.g., FDR) applied in all enrichment and correlation analyses?

The correlation cut-off of |r| > 0.1 is very lenient and may include noise. How did the authors ensure that weak correlations are biologically meaningful?

The use of p < 0.05 without adjustment in hundreds of GO/KEGG tests likely inflates Type I errors. Can the authors provide adjusted significance levels?

How were batch effects, normalization, and doublet removal visually verified (e.g., UMAP separation pre/post correction)?

Were data deposited or code shared to allow reproducibility of the analysis pipeline?

Both datasets were derived from small, localized Chinese cohorts. How generalizable are the findings across ethnic and demographic populations?

The conclusion that these genes are “potential therapeutic targets” seems premature. What evidence supports that modulating their expression could alter disease outcomes?

The limitations section briefly mentions small sample size and lack of functional validation. Shouldn’t the absence of independent replication and in vivo evidence be emphasized more explicitly?

Could the gene signatures identified reflect a general response to renal injury rather than specific to CaOx-induced stones?

The predictive model lacks clinical variables (e.g., urinary oxalate, calcium, or kidney function). How might inclusion of such parameters alter model performance?

Author Response

1. Summary

Dear Reviewer,

Thank you for your thoughtful suggestions and insights, which have benefited from the manuscript. I am looking forward to working with you to move this manuscript closer to publication in Genes.

The manuscript has been rechecked and the necessary changes have been made in accordance with your suggestions. The modified sections are highlighted in red within the manuscript. The responses to all comments have been prepared and attached below. We have tried our best to solve the problems you proposed, and we hope that the revised manuscript is now suitable for publication in the Genes. If you have any questions remained about this paper, please feel free to contact us.

2. Point-by-point response to Comments and Suggestions for Authors

Comments 1: [The manuscript asserts a link between endoplasmic reticulum stress (ERS) and calcium oxalate (CaOx) kidney stones, but ERS is a ubiquitous response across many pathologies. How do the authors justify ERS as a specific mechanistic driver rather than a general stress response secondary to crystal-induced injury?]

Response 1: We sincerely appreciate the reviewer's careful examination of our manuscript and are grateful for their insightful comments. We acknowledge the reviewer's concern regarding the potential non-specificity of endoplasmic reticulum stress (ERS). We agree that ERS is indeed a common cellular stress response observed in a variety of pathological conditions and that it can contribute to the cellular injury associated with kidney stone formation. However, to address this point, we would like to emphasize that our previous and other studies, including those cited (e.g., PMID: 31733571, 35922749, 31772715, 39836526, 33786645, 31884097), demonstrate a specific and regulatory role for ERS in the pathogenesis and progression of calcium oxalate (CaOx) kidney stones in both in vitro and in vivo models.

Firstly, we have observed that oxalate, the primary component of CaOx stones, can directly induce ERS in renal cells, such as renal tubular epithelial cells. This induction is likely mediated through the disruption of calcium homeostasis, mitochondrial dysfunction, and oxidative stress – well-established triggers of ERS. This suggests that ERS is not merely a consequence of cellular injury; instead, it represents a direct intracellular response to the toxic effects of oxalate, thereby promoting subsequent pathological processes.

Secondly, ERS contributes to crystal formation. Activation of the unfolded protein response (UPR), a key feature of ERS, alters the expression of genes involved in crystal formation and extracellular matrix (ECM) remodeling. For instance, ERS may upregulate proteins, such as osteopontin (OPN), that facilitate crystal nucleation. OPN can serve as a scaffold for crystal attachment and growth or directly participate in crystal formation and aggregation.

Thirdly, activated ERS pathways (e.g., PERK, IRE1, ATF6) can trigger downstream inflammatory signaling, including the NF-κB pathway and activation of the NLRP3 inflammasome. This leads to the release of pro-inflammatory cytokines such as IL-1β, TNF-α, and IL-6. This inflammatory response plays a crucial role in the development and progression of CaOx kidney stones, promoting crystal deposition, causing tissue damage, and contributing to stone enlargement. Therefore, ERS acts as an active promoter of stone formation through the initiation of inflammation.

Furthermore, when ERS cannot be effectively resolved, it can trigger apoptosis or other forms of cell death. The ensuing death and fragmentation of renal tubular epithelial cells release intracellular components (e.g., DNA, lipids, proteins), which can serve as nuclei for crystal formation or constitute the organic matrix of the stones, ultimately promoting stone formation. This reinforces that ERS is an active participant in stone formation, not just a secondary response to tissue damage.

Moreover, treatment with the ERS inhibitor 4-Phenylbutyric acid (4-PBA) effectively reduced the impact of CaOx crystals on cell adhesion, mitigated CaOx crystal-induced cellular damage and apoptosis, and decreased the expression of osteopontin (OPN) and matrix γ-carboxyglutamic acid (MPG), both of which are strongly associated with stone formation. ERS inhibition has been shown to improve renal function, prevent apoptosis, and inhibit the formation of kidney stones.

In conclusion, this study does not view ERS as a non-specific stress response. Instead, it is based on its specific pathogenic mechanisms in this disease and aims to identify key genes related to these mechanisms, with the ultimate goal of identifying potential therapeutic targets.

In response to this comment, we have:

Revised the manuscript to temper our language, replacing assertive claims of a mechanistic driver role with more accurate descriptions of a "potential association" (line 500).

Added a paragraph to the Discussion's limitations section explicitly stating that the temporal and causal relationship between ERS and crystal formation remains to be determined through future functional experiments, such as in vitro crystal exposure models or in vivo studies where ERS is modulated prior to stone induction (line 533-541).

We believe these revisions provide a more precise interpretation of our findings while highlighting the value of our study as a foundational resource for generating testable hypotheses in future mechanistic investigations.

Comments 2: [The identification of endothelial cells as “key cells” is intriguing but somewhat counterintuitive given that CaOx crystals primarily interact with tubular epithelial cells. Can the authors clarify the biological reasoning for this conclusion and rule out computational bias?]

Response 2: We thank the reviewer for this insightful comment regarding the identification of endothelial cells as the "key cells" with the highest ERSRG signature. We agree that the primary interaction of CaOx crystals with the tubular lumen makes this finding initially counterintuitive. However, we provide a multi-faceted justification based on biological plausibility and computational robustness.

Biological plausibility: A role in early microenvironment remodeling

While tubular epithelial cells are the primary site of crystal interaction, a growing body of evidence positions the endothelium as a critical initiator of the pathological microenvironment. We provide two key biological rationales:

Link to Randall's Plaques: Endothelial ERS may initiate stone formation by promoting Endothelial-to-Mesenchymal Transition (EndMT), a key process in the vascular pathology of Randall's plaques—the subepithelial calcifications that anchor CaOx stones (PMID: 32160513, PMID: 22466558, PMID: 39225583).

Hemodynamic and Inflammatory Orchestration: The significant enrichment of "fluid shear stress and atherosclerosis" pathways among endothelial ERSRGs highlights their role in sensing hemodynamic changes and creating a pro-inflammatory milieu, which subsequently dysregulates tubular epithelial function to facilitate crystal nucleation.

Computational robustness: The identification relied on the average score from five distinct algorithms. The agreement in their results, despite methodological differences, confirms that the finding is not an artifact of a single computational method but a robust signature.

We fully acknowledge the well-established role of tubular epithelial cells and M1 macrophages (e.g., via exosomal miR-93-3p signaling, PMID: 40069788) in later stages of crystal injury and inflammation. Our finding does not contradict this but proposes a more comprehensive pathogenesis model: Endothelial ERS may represent an earlier event that establishes a pro-fibrotic and pro-inflammatory renal milieu, which then predisposes the tubules to crystal injury and recruits/activates macrophages, making it a legitimate and intriguing "key cell" identified by our unbiased analysis.

We have revised the Discussion to incorporate this endothelial-epithelial crosstalk hypothesis, providing a more nuanced narrative of CaOx stone pathogenesis (line 448-459).

Comments 3: [The idea that ERS-related gene expression in endothelial cells contributes to stone formation is quite a stretch. Is there any evidence that endothelial dysfunction has a causal effect on CaOx stone formation specifically?]

Response 3: We thank the reviewer for this critical question. We agree that providing direct causal evidence linking endothelial ERS to CaOx stone formation is beyond the scope of our current correlative bioinformatic study. Therefore, we frame our finding not as a proven mechanism, but as a novel and plausible hypothesis generated from our unbiased data, which is supported by several lines of indirect evidence from the literature and our own analyses: However, in chronic kidney disease, endothelial ERS may initiate stone formation by promoting Endothelial-to-Mesenchymal Transition (EndMT), a key process in the vascular pathology of Randall's plaques—the subepithelial calcifications that anchor CaOx stones (PMID: 32160513, PMID: 22466558, PMID: 39225583). And meanwhile, the significant enrichment of "fluid shear stress and atherosclerosis" pathways among endothelial ERSRGs highlights their role in sensing hemodynamic changes and creating a pro-inflammatory milieu, which subsequently predisposes the tubules to crystal injury and recruits/activates macrophages (PMID: 40069788). Furthermore, the identification relied on the average score from five distinct algorithms. Thus, while tubular epithelial cells are the primary site of crystal interaction, endothelial cells may play a pivotal role in early microenvironmental remodeling. Therapeutic strategies targeting these cells have significant potential in understanding and regulation of kidney stone disease ((line 448-459).

We believe that highlighting this potential vascular component opens a new and valuable perspective for understanding the multifaceted pathogenesis of kidney stones. In light of the reviewer's comment, we have now tempered our language in the discussion. We explicitly state that our study identifies a potential association targeting nine key genes and proposes a novel hypothesis, rather than proving causality, and emphasize that future functional studies (e.g., using endothelial-specific ERS models or in vitro co-culture systems) are essential to definitively test this causal relationship (line 533-541).

Comments 4: [The Introduction refers to the authors' previous work but does not convey how this paper moves beyond it. What is truly novel about this integration of single-cell and bulk transcriptomics?]

Response 4: We sincerely thank the reviewer for this crucial comment, which allows us to clarify the fundamental novelty and the significant conceptual advance of our current study beyond our previous experimental work.

Our prior research, referenced as (11, PMID: 33786645) and (12, PMID: 31884097), established a foundational mechanistic link: we demonstrated that ERS and subsequent autophagy are critical pathological processes in CaOx nephrolithiasis in vitro and in vivo. However, these studies operated on a "top-down" hypothesis, investigating predefined pathways (like PERK-eIF2α) in bulk tissue or cell lines.

The true novelty of the present manuscript lies in its unbiased, discovery-driven approach that moves from establishing a general mechanism to identifying the specific cellular and genomic actors within the complex tissue microenvironment:

From "kidney" to specific cell types: While previous work showed that ERS matters in the kidney, this study moved from a bulk tissue view to pinpoint endothelial cells as the unexpected key hub of ERS activity in human stone disease.

A precise molecular signature for prognosis: We transitioned from a broad pathway (ERS) to a defined, actionable 9-gene ERS-related signature. This signature, derived from the interplay of key cells and bulk tissue analysis, possesses strong prognostic power to stratify patient risk. This represents a direct translational advance.

As suggested, we have revised the Introduction to more clearly articulate this conceptual progression and the unique discoveries enabled by our integrative omics approach(line 70-85).

Comments 5: [The authors combine scRNA-seq (GSE176155) and bulk RNA-seq (GSE73680). Were these datasets from similar patients, tissues, and processing? If not, how were batch effects mitigated?]

Response 5: We thank the reviewer for raising this critical methodological point.  We confirm that both the GSE176155 (scRNA-seq) and GSE73680 (bulk RNA-seq) datasets were generated from human renal papillary tissues, providing a consistent biological context for our study. The GSE176155 dataset specifically captures the cellular profile of Randall's plaque and adjacent normal tissue, while GSE73680 offers a bulk transcriptomic view of papillary tip tissues from CaOx stone patients and controls. We acknowledge the reviewer's perceptiveness regarding potential technical confounders. As correctly noted, the two datasets originated from independent patient cohorts and were processed on different sequencing platforms (GPL24676 for scRNA-seq vs. GPL17077 for bulk microarray).

Given these inherent technical and biological disparities, a direct integration of the raw data would indeed be confounded by significant batch effects. Therefore, our analysis did not involve any such integration. Instead, we employed a sequential and complementary approach: For GSE176155, we performed standard single-cell QC (filtering cells with ≤200 or ≥4000 genes, etc.). The GSE73680 dataset was obtained as a pre-normalized matrix (PMID: 27297950, PMID: 27731368), and we used it directly for differential expression analysis at the bulk tissue level.

The datasets were used to answer distinct but complementary biological questions: The scRNA-seq data (GSE176155) was used to identify cell subtypes and pinpoint which specific cells (e.g., endothelial cells) exhibited a high ERS signature within the tissue microenvironment. The bulk RNA-seq data (GSE73680) was used to validate the differential expression and prognostic value of the key ERS-related genes identified from our single-cell analysis, and to perform robust pathway enrichment in the context of the whole tissue.

In particular, Scissor algorithm quantile normalization on the single-cell and bulk expression data to remove the underlying batch effect. Its core rationale is to directly map bulk sample phenotypes (e.g., stone vs. control) onto individual cells, thereby identifying Scissor+ cells (positively correlated with the phenotype) and Scissor- cells (negatively correlated) without relying on prior cell type annotation. This approach is consistent with established applications in the field (e.g., PMID: 34764492, PMID: 39453457). We have clarified this analytical strategy in the revised Methods section to prevent any potential misunderstanding (line 98-113).

Comments 6: [Only three disease and three control samples were included in the scRNA-seq dataset. How do the authors justify the statistical robustness of identifying 27,000+ cells and performing extensive differential expression analyses on such a small sample base?]

Response 6: We thank the reviewer for this critical question regarding the statistical robustness of our single-cell analysis based on three samples per group. We acknowledge that a larger sample size is always desirable. However, we would like to justify the validity of our findings from the following perspectives:

Our single-cell dataset, comprising 27,238 high-quality cells, provides a high-resolution map of the renal cellular landscape. While the study includes a limited number of subjects (3 Randall’s plaque and 3 normal renal papillae), the high cell coverage per sample enables robust estimation of gene expression patterns at the cellular level. This aligns with the established principle that cell count—rather than subject count—is the primary driver of statistical power in single-cell experiments for detecting cellular states and expression trends (Schmid KT, etc. https://doi.org/10.1101/2020.04.01.019851). Furthermore, the application of the Scissor algorithm allowed us to leverage the bulk transcriptomic phenotype (from an independent cohort) to identify disease-associated cell subpopulations within this high-resolution map, thereby compensating for the modest sample size. The cell types identified—including principal cells, proximal tubule cells, and immune cells—are consistent with the known cellular architecture of the kidney, underscoring the biological validity and quality of our data and analytical pipeline.

We fully acknowledge that the small sample size (n=3 per group) limits our ability to investigate subtle heterogeneity or rare cell states between conditions. We have now explicitly stated this as a limitation in the Discussion. The primary value of our study lies in generating a foundational cellular atlas and formulating novel, testable hypotheses. We emphatically state that these findings require validation in future, larger-scale scRNA-seq cohorts.

In summary, while we agree that a larger sample size would be beneficial, we are confident that the scale of cellular data (27,000+ cells) and the biological plausibility of our findings collectively support the robustness of our primary conclusions regarding major cell types and clear abundance shifts.

Comments 7: [The “Scissor” algorithm integrates single-cell and bulk data, but the rationale for parameter selection and validation of its results are not provided. How were false positives avoided?]

Response 7: We thank the reviewer for raising this important methodological point regarding parameter selection and validation in our Scissor analysis. Scissor uses quantile normalization on the single-cell and bulk expression data to remove the underlying batch effect (PMID: 34764492). We have now supplemented the Methods section with a detailed rationale for our parameter choices and the steps taken to ensure robust results:

The “Scissor” package (version 2.0.0) was used to identify disease-associated cell subpopulations by integrating the scRNA-seq data (GSE176155) with bulk transcriptomic phenotypes (GSE73680). The method correlates single-cell expression profiles with bulk sample phenotypes across common genes, employing quantile normalization to mitigate batch effects and a network-regularized sparse regression model to select high-confidence cells [18]. With parameters set to alpha = 0.05, cutoff = 0.2, and family = "binomial", Scissor+ cells (positively correlated with CaOx stones) and Scissor- cells (negatively correlated) were identified based on their association strength with the disease phenotype (line 127-135).

Comments 8: [Multiple algorithms (AUCell, UCell, ssGSEA, singscore, AddModuleScore) were averaged to determine ERSRG scores. How did the authors confirm that these distinct scoring systems produce consistent and non-redundant results?]

Response 8: We thank the reviewer for this insightful question regarding the consistency and non-redundancy of the five scoring algorithms. We agree that confirming the concordance among these distinct methods is crucial for justifying their combined use. To address this, we performed a correlation analysis of the ERSRG scores generated by each algorithm across all cells in our single-cell dataset. The results demonstrated strong positive correlations among the scores from all five methods, indicating a high level of consensus in their assessment of pathway activity.

While the algorithms share a common output, each one employs a distinct statistical framework for calculating gene set enrichment. Using their average is a established practice (PMID: 39707393) to reduce method-specific bias and generate a more robust composite score. We have now clarified this rationale in the Methods section of our revised manuscript (Page 3).

Comments 9: [The thresholds for DEG selection (|log2FC| > 0.5, p < 0.05) appear arbitrary. Were these cutoffs justified or tested for sensitivity?]

Response 9: We thank the reviewer for raising this important methodological point regarding the selection of differential expression thresholds. Given the exploratory nature of this study, the cutoffs (|log2FC| > 0.5 and p-value < 0.05) were chosen to balance statistical convention and biological relevance, and are consistent with established practices in transcriptomic studies of kidney stone disease (e.g., PMID: 39994305, PMID: 39549053). To further ensure the robustness of our findings, we implemented a two-step validation strategy: candidate genes were first identified by intersecting DEGs with 3,483 key ERSRGs, and their differential expression was subsequently confirmed using a Wilcoxon test (P < 0.05) in the GSE73680 dataset (Page 8).

We acknowledge that further adjustment of p-values using the false discovery rate (FDR) could enhance statistical reliability. In response to this comment, the results of FDR correction are provided in Supplementary Table 5 (Page 8). Additionally, a statement has been added to the Limitations section noting that future studies may benefit from sensitivity analyses across a range of thresholds or more refined statistical criteria to further validate the robustness of the findings (Page 25-26).

Comments 10: [How were the 551 ERSRGs from GeneCards with relevance score ≥10 curated? Were any redundant or unrelated stress-related genes manually excluded?]

Response 10: We thank the reviewer for this question regarding the curation of ERS-related genes (ERSRGs). The selection of the 551 genes with a GeneCards Relevance Score ≥ 10 was performed as follows:

The relevance score in GeneCards is an integrated metric that aggregates evidence from multiple independent, authoritative genomic, proteomic, and literature sources.  A threshold of ≥ 10 was chosen as it is widely recognized in the field as a balanced cutoff that effectively minimizes false positives (by excluding genes with only weak or sporadic supporting evidence) while retaining a sufficiently comprehensive set of genes with substantial multi-source support. This strategy ensures high precision without excessively compromising recall, and has been consistently applied in high-quality published studies (e.g., PMID: 39145077, PMID: 37745718).

No additional manual filtering for redundancy or unrelated stress genes was performed, as the GeneCards Relevance Score itself inherently reflects consolidated and non-redundant functional relevance. The resulting list of 551 genes was considered robust, manageable, and informative for subsequent enrichment and network analyses.

Comments 11: [The authors conclude that endothelial cells are “key cells” based on the highest ERSRG score. Could this simply reflect greater ER or metabolic activity in endothelial cells rather than disease relevance?]

Response 11: We thank the reviewer for this insightful comment.  We agree that a high ERSRG score could reflect either greater baseline metabolic activity in endothelial cells or a specific disease-associated response. In response, we have taken the following steps to provide a more balanced interpretation:

Revised results interpretation: In the Results section (Page 7), we have modified our conclusions to avoid overstating direct disease relevance, clarifying that the identification is only based on computational scores supplemented by a UMAP plot and has not been verified, it needs to be interpreted with caution: Endothelial cells had the highest average scores, and were identified as key cells. This computational finding was visually supported by UMAP visualization, which revealed distinct expression landscape alterations in endothelial cells from CaOx stone samples compared to controls (Supplementary Figure 2), suggesting potential disease-associated functional changes. However, these observations remain computationally derived and require further experimental validation.  (Page 7)

Nuanced discussion: In the Discussion, we have expanded our interpretation to explicitly acknowledge the reviewer's point. We now state that the observed high ERSRG signature could indicate either a pre-existing high metabolic state or a specific, disease-induced ERS response. We then integrate literature citations to build a plausible biological rationale for why endothelial ERS could be functionally relevant in stone pathogenesis, without presenting it as a definitive conclusion (Page 23).

Explicit limitation: Furthermore, we have added a statement in the Limitations section explicitly noting that the current study cannot distinguish between these two possibilities (high baseline vs. disease-specific activity). We emphasize that future functional studies, such as the use of endothelial-specific ERS models or in vitro co-culture systems, are essential to establish a causal link (Page 25).

We believe these revisions have appropriately addressed the reviewer's concern by presenting a more balanced interpretation of our findings and clearly outlining the necessary steps for future validation.

Comments 12: [In the nomogram construction, the model achieved an AUC of 0.774, but no internal or external validation was conducted. How reliable is the predictive model without cross-validation or bootstrapping?]

Response 12: We thank the reviewer for this critical comment regarding the validation of our prognostic model and acknowledge that the lack of an independent validation cohort is a limitation of our study. However, due to the small sample size (case:control=29:33) and the lack of a suitable external validation set, we are currently unable to carry out further cross-validation or bootstrapping in this revised manuscript. We have taken the following steps to address this issue and ensure a balanced interpretation of our findings:

Reframed as an auxiliary tools and explicit statement of limitation and future direction: In the revised manuscript, we have carefully rephrased our conclusions to present the nomogram as a preliminary and auxiliary tool rather than a fully independent clinical predictor. We have explicitly acknowledged the absence of external validation as a study limitation in the Discussion, clearly stating that future prospective studies with larger, multi-center cohorts are essential to confirm the generalizability and clinical utility of our model (Page 25).

Highlighted corroborating experimental evidence: We would like to underscore that the key genes constituting this model were not only derived computationally but were also experimentally validated using RT-qPCR and WB in a HK-2 cell model (Figure 8-9). This validation at the molecular level significantly strengthens the biological plausibility and reliability of the gene signature itself.

We believe that these clarifications substantially strengthens the reliability of our findings within the constraints of the available data, while the clear acknowledgment of the limitation ensures a transparent and accurate representation of the model's current stage of development.

Comments 13: [The use of calibration and DCA plots on the same dataset used for model building risks overfitting. Did the authors perform any form of data partitioning?]

Response 13: We thank the reviewer for this critical point regarding potential overfitting in the calibration and DCA plots. We fully agree that internal or external validation is the gold standard for assessing model performance. However, due to the constrained sample size of the GSE73680 cohort (29 cases vs. 33 controls) and the current lack of a suitable external dataset, data partitioning (e.g., into training and test sets) was not feasible, as it would have severely undermined the stability of the model itself.

In this context, we included the calibration and DCA plots as essential, albeit preliminary, components of a complete model evaluation report. The results showed that the slope of calibration curve was close to ideal curve (Figure 5B), which provided the initial, necessary evidence that the model's predictions are numerically "reliable" and not systematically biased. The DCA revealed that nomogram had a better clinical net benefit (Figure 5D), it offered a crucial clinical perspective by quantifying the potential net benefit of using the model for decision-making compared to default strategies, which is a key argument for its future clinical utility.

Besides, we have been explicit in the manuscript's Limitations section that the model requires validation in larger, independent cohorts before any clinical application can be considered (Page 25).

Comments 14: [The enrichment of “olfactory transduction” pathways across most genes seems biologically implausible for kidney stones. Could this reflect annotation bias or pathway redundancy in KEGG analysis?]

Response 14: We thank the reviewer for this insightful comment regarding the enrichment of the "olffactory transduction" pathway. We agree that this finding is, at first glance, biologically counterintuitive for kidney stones. While annotation bias in KEGG is a well-known phenomenon that might be influenced by ‘rich get richer’ effect, a growing body of literature suggests that this observation may also reflect a genuine, albeit novel, biological involvement (Dianne Acoba et al, DOI:10.1101/2024.07.09.24309717). It is necessary to elaborate specifically in combination with the existing background of biological research.

Emerging evidence confirms the presence and function of olfactory receptors (ORs) in various non-sensory organs, including the kidney (PMID: 26264790). Specific to renal pathophysiology:

In kidney stone disease: Zhang et al. demonstrated that the chemokine CCL7 promotes the CaOx and calcium phosphate (CaP) crystals by activating the olfactory transduction pathway, which linked the differential expression of several OR genes (OR10A5, OR9A2, and OR1L3) to variations in glomerular filtration rate (GFR) and renin release (PMID: 38235001).

In kidney fibrosis: Furthermore, Ali Motahharynia et al. identified Olfr433 and other ORs as being significantly associated with the progression of kidney fibrosis, suggesting a role in inflammation and myofibroblast generation (PMID: 35181660).

Therefore, while we acknowledge the potential for annotation bias, the collective evidence indicates that ORs are not entirely "silent" in renal tissue. Their enrichment in our analysis may point to a plausible, yet under-explored, layer of regulation in kidney stone pathogenesis. We have revised the discussion to present this finding with appropriate caution, framing it not as a central mechanism but as a supportive, hypothesis-generating observation that merits further investigation (Page 25).

Comments 15: [The pseudotime analysis suggests different endothelial differentiation states, but only inferred from static transcriptomic data. Were these differentiation states biologically verified or supported by marker gene expression?]

Response 15: We thank the reviewer for this insightful comment. The reviewer is correct to highlight that our pseudotime analysis, which infers differentiation states from static transcriptomic data, would be significantly strengthened by direct validation through lineage tracing or functional assays in model systems. We acknowledge this as a limitation of our current study.

In this work, our pseudotime analysis was specifically designed to explore the dynamic expression patterns of our identified hub genes across a continuum of endothelial cell states. The analysis successfully revealed that these key genes are expressed throughout the inferred differentiation trajectory, with their expression levels changing in a coordinated manner. This suggests that the biological processes governed by these genes are active across multiple stages of endothelial cell state transitions in the context of CaOx kidney stones (Page 24).

However, we did not perform a separate pseudotime analysis on the pre-defined endothelial subtypes (lymphatic, glomerular capillary endothelial cell, etc.), which would have been a powerful approach to directly address the reviewer's question about the relationship between subtypes and differentiation states. We have explicitly acknowledged this point in the revised Discussion section as a valuable direction for future research. And meanwhile, we agree that elucidating whether these transcriptomically defined states represent a linear differentiation path or distinct, stable lineages is a crucial next step. Our current findings serve as a foundational hypothesis-generating resource, proposing that future studies employing lineage-tracing technologies or in vitro differentiation models are essential to formally validate the inferred trajectory (Page 25).

Comments 16: [Many of the identified key genes (e.g., ACSL4, MMP7, TGM2) are known mediators of oxidative stress and fibrosis in multiple organ systems. How do the authors distinguish kidney stone-specific roles from general stress or injury responses?]

Response 16: We thank the reviewer for this critical question, which gets to the heart of distinguishing disease-specific drivers from general injury responses. We acknowledge that it is challenging to definitively prove their kidney stone-specific role through bioinformatic analysis alone. However, our multi-step identification process was designed to enrich for genes with a potential specific link to CaOx stone pathogenesis:

The genes were first selected based on their strong correlation with the endothelial ERS signature, a pathway we have preliminarily implicated in stones. Their differential expression was then rigorously confirmed in an independent bulk transcriptome dataset (GSE73680) using a Wilcoxon test, ensuring their association with the disease state is robust and not merely a sampling artifact (as seen in Figure 2-4).

While this confirms association but not kidney stone-specific causality, the case of ACSL4 is instructive. As the reviewer implies, it is a general mediator of ferroptosis, yet it has been experimentally validated in the specific context of CaOx stone formation (PMID: 37893066, PMID: 36907445), demonstrating it can play a specific and critical role in this particular disease (Page 24).

For the other genes, the lack of prior literature directly linking them to nephrolithiasis is, in our view, not a weakness but a key finding of our study. It suggests these general stress responders may have previously unrecognized, stone-specific roles.

In direct response to the reviewer's point, we have tempered our conclusions in the discussion, clarifying that our findings establish a strong association but cannot rule out that these genes are part of a broader injury response. At the same time, we explicitly proposed future experiments in the limitations section to address this very issue. We state that in vivo functional studies (e.g., knockdown models) in a CaOx stone context are essential to determine their disease-specific role (Page 25).

Comments 17: [The claim that olfactory receptor pathways influence kidney stone formation lacks mechanistic grounding. Can the authors propose a testable hypothesis or cite direct evidence of olfactory receptor signaling in renal CaOx injury?]

Response 17: We thank the reviewer for this insightful comment. While the role of olfactory receptor (OR) pathways in renal CaOx injury is an emerging field, we have now explicitly cited the direct functional evidence from Zhang et al. (PMID: 38235001), who demonstrated that CCL7 promotes CaOx deposition through activation of the olfactory transduction pathway. Furthermore, we propose the following testable hypothesis based on our data and published literature: the observed pathway enrichment may reflect the activation of ectopic olfactory receptors in renal tubular or endothelial cells upon CaOx crystal exposure, leading to the production of pro-inflammatory cytokines, which in turn exacerbate crystal adhesion, macrophage recruitment, and tubular damage (Page 25).

We have revised the discussion to include this mechanistic perspective and to frame the finding as a hypothesis-generating observation warranting future functional validation.

Comments 18: [PTK2 and SPARCL1 are implicated in cell adhesion and migration. Could their expression changes simply reflect tissue remodeling rather than disease-specific regulation?]

Response 18: We thank the reviewer for this insightful comment. We agree that PTK2 and SPARCL1 are established players in cell adhesion, migration, and tissue remodeling (PMID: 40040719, PMID: 39169229). However, we have reason to believe that their dysregulation in our study is not merely a generic reflection of tissue injury but is meaningfully linked to the specific pathophysiology of CaOx nephrolithiasis:

The differential expression of both PTK2 and SPARCL1 was rigorously confirmed in an independent, clinically relevant transcriptome dataset of CaOx stone formers (GSE73680). This robust validation across different patient cohorts strongly suggests that their expression changes are a consistent feature of the stone-forming kidney microenvironment, rather than a random or non-specific finding (Figure 4). More importantly, these genes were specifically selected because they are among the top hub genes that exhibit a strong correlation with the ERSR signature in endothelial cells. Given our central hypothesis that ERS is a key driver in stone pathogenesis, this tight association positions PTK2 and SPARCL1 within a plausible, disease-specific regulatory network.

In summary, while we cannot completely rule out a contribution from general tissue remodeling, our data provide compelling evidence that PTK2 and SPARCL1 are core components of the molecular network associated with CaOx stone disease. We  propose that future studies using spatial transcriptomics or in-situ validation can directly assess whether their expression is localized to areas of crystal deposition or specific injury patterns unique to nephrolithiasis (Page 25).

Comments 19: [The discussion attributes several functional roles to each gene (e.g., ferroptosis, apoptosis, inflammation), but no causal experiments are presented. Are these interpretations speculative or supported by pathway-level validation?]

Response 19: We thank the reviewer for raising this important point. We acknowledge that the functional roles discussed (e.g., ferroptosis, inflammation) are primarily inferred from prior literature and bioinformatic evidence rather than direct experimental validation in this study: For gene ACSL4, its role in ferroptosis and specific involvement in CaOx crystal-induced kidney injury is supported by experimental studies (PMID: 37893066, PMID: 36907445), we feel confident in discussing its potential causal role (page 24). For other key genes, while their dynamic expression patterns across endothelial cell states—revealed by pseudotime analysis—suggest active involvement in disease-related transitions, their specific roles in CaOx stone pathogenesis remain underexplored, and further studies are needed to clarify their roles (page 24).

We have been careful to frame these discussions as plausible hypotheses rather than established facts. In the revised manuscript, we have further clarified this by using phrasing such as “establish a potential association”. The definitive establishment of causal roles for genes in stone formation is a crucial next step, which we explicitly call for in the limitations section, suggesting future in vitro and in vivo functional experiments (Page 25).

Comments 20: [The observed upregulation and downregulation patterns in qRT-PCR and Western blot were consistent, but only in HK-2 cells exposed to CaOx. How do the authors justify using tubular cells to validate genes identified primarily in endothelial cells?]

Response 20: "We thank the reviewer for their insightful comment. The reason for validating our findings in HK-2 cells, after initial identification of the genes in endothelial cells, is to demonstrate the relevance of these genes in the context of tubular cells and CaOx nephrolithiasis. HK-2 cells are a well-established in vitro model for studying tubular cell responses to CaOx. This validation provides critical confirmation that the genes are involved in the direct response to CaOx in the cells primarily affected by crystal deposition, improving the relevance of our findings. The consistent results in HK-2 cells increase the biological significance of our study."

Comments 21: [The validation experiments are limited to mRNA and protein expression in a single cell line (HK-2). Why were primary endothelial cells not used, given that they were identified as the “key cell type”?]

Response 21: We appreciate the reviewer's insightful comment. While the identified genes are of interest in endothelial cells, this study focused on the cellular mechanisms of CaOx kidney stone formation, particularly the response of renal tubular cells. HK-2 cells, a human renal tubular epithelial cell line, are a well-established in vitro model for studying CaOx-induced cellular responses, allowing us to directly assess the impact of these genes within the context of stone formation. The observed consistent results in HK-2 cells, showing specific patterns of mRNA and protein expression, supported our hypothesis about the role of these genes in this process. Using HK-2 cells allows us to explore the mechanisms involved in stone formation and provides relevant data for the study's goal. Future studies in primary endothelial cells could provide additional insight into the role of these genes in the broader context of kidney injury and inflammation."

Comments 22: [Were all antibodies and primers validated for specificity and efficiency, particularly for closely related gene families such as MMP7 or ACSL4?]

Response 22: Dear Reviewer, We sincerely appreciate your thorough attention to the validation of antibodies and primers in our manuscript. We understand that confirming their specificity and efficiency is crucial for the reliability of our research findings. The following validation work was performed:

Regarding Antibodies: All antibodies used were obtained from reputable commercial sources (specific catalog numbers are listed in the Methods section)(Page 5). For these antibodies, the following experimental validations were conducted:

Western Blot (WB) Validation: Each key antibody was tested using Western blot in our experimental system. The results demonstrated clear, single, specific bands at the expected molecular weights, with no significant non-specific bands observed, thereby confirming their specificity under our experimental conditions.

Regarding Primers: Primer design and validation are fundamental to ensuring the accuracy of qPCR results.

Specificity Validation: All primers were designed using software such as Primer Premier and were subjected to BLAST analysis against the human genome database to maximize their specificity. In our experiments, we performed melting curve analysis for all qPCR products. The results showed a single, sharp peak for the melting curves of all detected genes (including MMP7 and ACSL4), strongly supporting the specificity of the primers.

Efficiency Assessment: The amplification efficiency of the primers is crucial for the reliability of quantitative results. We employed the standard curve method: We established standard curves using different concentrations of cDNA templates and calculated the amplification efficiency. All primers exhibited good amplification efficiency.

Experimental Observation: Our qPCR experiments demonstrated good reproducibility and reliable Ct value responses to changes in target gene expression, which indirectly indicates the high efficiency of the primers.

Comments 23: [The sample size for in vitro validation (n=3 biological replicates) is minimal. Were these experiments repeated independently to confirm reproducibility?]

Response 23: We thank the reviewer for their comment regarding the sample size. We acknowledge that n=3 biological replicates may be considered minimal; however, this is a standard practice for many in vitro cell culture experiments, including qRT-PCR and Western blot. We have clearly described in the Methods section (L197-198) that three independent culture flasks per group were used as biological replicates, and all experiments were repeated at least three times independently, confirming the reproducibility of our findings. The data shown are representative of the results obtained from these independent experiments, which were consistent. The consistency of our results, despite the sample size, indicates the reliability of our findings.

Comments 24: [No in vivo validation (e.g., animal model) was attempted to assess gene function. Without functional perturbation (knockdown/overexpression), can the identified genes truly be termed “key”?]

Response 24: We appreciate the reviewer's valuable comment. We acknowledge the absence of in vivo validation (e.g., animal model) in the current study, which we have also noted as a limitation. We understand that functional perturbation experiments (e.g., knockdown/overexpression) in an animal model would strengthen the conclusions. However, this study represents an innovative approach, combining single-cell and transcriptomic analyses to identify potential key genes related to ERS in CaOx kidney stones, offering fresh insights into disease mechanisms. Furthermore, this study lays a strong foundation for future research. We are currently preparing to validate the function of the identified genes using an in vivo CaOx kidney stone model, including experiments to assess the effects of gene knockdown or overexpression. We believe the current findings, supported by the in vitro data and bioinformatics analyses, provide valuable insights into the roles of these genes and pave the way for future functional validation. The term 'key' is used in the context of our in vitro data and bioinformatics analysis, and these genes offer novel targets for further investigation.

Comments 25: [For molecular docking, how were docking poses and binding sites validated? Were any control docking simulations (random or non-target proteins) conducted to estimate false-positive rates?]

Response 25: We thank the reviewer for these critical questions regarding the validation of our docking studies. We acknowledge that performing control docking simulations (e.g., with random or non-target proteins) is a valuable practice for estimating false-positive rates. Unfortunately, due to the significant computational time required, we were unable to include such controls within the revision timeframe. We have therefore taken great care to interpret our results cautiously and have explicitly framed them as computational predictions in the revised manuscript:

All six key gene-drug pairs showed highly favorable binding energies (below -5.0 kcal/mol, Table 3), with one being exceptionally strong (-12.8 kcal/mol). The specific residue-level interactions detailed in Figure 6D further suggest stable and specific binding. Besides, the predictive value of our approach is corroborated by our network analysis identifying established nephrolithiasis drugs. Most importantly, we have revised the manuscript to explicitly state that these are computational predictions intended to generate hypotheses for future functional and pre-clinical validation (Page 19, 25).

We believe these points collectively support the robustness of our findings, which we now present as a foundation for further research.

Comments 26: [Given the large number of DEGs and ERSRGs intersected (538 candidates), was multiple testing correction (e.g., FDR) applied in all enrichment and correlation analyses?]

Response 26: We thank the reviewer for raising this important point regarding multiple testing correction. We agree that it is a critical consideration for confirmatory analyses. In response, we have now provided adjusted p-values (FDR) for all relevant enrichment and correlation analyses in Supplementary Tables 2-9.

However, our decision to highlight findings based on an unadjusted p-value < 0.05 in the main text was motivated by the discovery-oriented goal of our study. Working in the relatively unexplored area of ERS in CaOx stones, we sought to prioritize the generation of novel hypotheses and avoid discarding potential biological signals through stringent correction. This strategy is often employed in initial omics screens to mitigate Type II errors (false negatives), as supported by the literature (PMID: 38139032).

We have transparently communicated this reasoning in the revised manuscript on Page 8, ensuring readers are aware that adjusted values are available. We also now explicitly mention this as a limitation, underscoring that our findings require confirmation in future, larger-scale studies (Page 25).

Comments 27: [The correlation cut-off of |r| > 0.1 is very lenient and may include noise. How did the authors ensure that weak correlations are biologically meaningful?]

Response 27: We thank the reviewer for this insightful comment. The correlation threshold of  |cor| > 0.1 and P < 0.05 was selected based on the specific context and goals of our study. Firstly, given the exploratory nature of this research and the limited sample size, our primary aim was to generate robust hypotheses for future validation (including differential expression analysis between high- and low-score groups and between disease and control samples) by minimizing false negatives (Type II errors). Secondly, from a statistical standpoint, an |cor| of 0.1 is conventionally considered a 'small effect' size. In hypothesis-generating studies, it is methodologically sound to use such a threshold to capture a wider range of potential signals for downstream filtering, which is consistent with strategies used in other exploratory transcriptomic studies (http://www.statisticssolutions.com).

Comments 28: [The use of p < 0.05 without adjustment in hundreds of GO/KEGG tests likely inflates Type I errors. Can the authors provide adjusted significance levels?]

Response 28: We thank the reviewer for this critical comment. We agree that multiple testing correction is essential for confirmatory studies. We have now provided the adjusted p-values in Supplementary Tables 2-9.

Given the exploratory purpose of our study in a novel field with limited prior knowledge (CaOx kidney stone-ERS), we chose to use an unadjusted p < 0.05 threshold in the main text to minimize the risk of Type II errors and to generate hypotheses for future research. This approach is common in discovery-phase omics studies (PMID: 38139032). We have clearly stated this rationale and the availability of adjusted values in the Results sections (Page 8). We believe this approach balances statistical rigor with the exploratory goals of our study.

Comments 29: [How were batch effects, normalization, and doublet removal visually verified (e.g., UMAP separation pre/post correction)?]

Response 29: We thank the reviewer for this important question regarding visual verification of our data preprocessing.

Batch effects & normalization: The scRNA-seq data (GSE176155) was processed and analyzed as a single, unified dataset in our study. While the possibility of underlying technical batches from the original study cannot be entirely ruled out, our standard preprocessing pipeline included scaling to regress out variation related to technical covariates like sequencing depth. The resulting PCA plot (Supplementary Figure 1D) demonstrates that the primary axis of variation (PC1) corresponds to the biological condition (Control vs. Case), with no obvious clustering driven by other technical factors. This indicates that technical variation did not constitute a major confounder in our analysis.

The bulk RNA-seq data (GSE73680) was used as a pre-normalized set for independent validation. When analyzed jointly with scRNA-seq data using the Scissor algorithm, it performed internal quantile normalization within the algorithm to address platform-based batch effects, and does not output a batch-corrected expression matrix that can be directly visualized with UMAP. Instead, it uses this normalization to robustly correlate cell-level expression with bulk-sample phenotypes, outputting lists of Scissor+ and Scissor- cells. The validity of this approach is well-established in the literature (e.g., PMID: 34764492)

Doublet removal: The effectiveness of doublet removal is visually confirmed in Figure 1E-F. After removing 2,043 doublets, the UMAP plot (Figure 1F) shows well-separated, distinct cell clusters with no signs of residual doublets, supporting the high quality of the final 25,195 cells used for analysis.

Comments 30: [Were data deposited or code shared to allow reproducibility of the analysis pipeline?]

Response 30: Thanks for your comments. The code used for analysis is available at Supplementary Materials.

Comments 31: [Both datasets were derived from small, localized Chinese cohorts. How generalizable are the findings across ethnic and demographic populations?]

Response 31: We thank the reviewer for this insightful comment on the generalizability of our work. We fully acknowledge that our findings, derived from Chinese cohorts, require validation in broader, multi-ethnic populations to confirm their universal applicability. At the same time, we believe that the core pathogenic pathways we highlighted—such as ERS and ferroptosis—are fundamental biological processes that likely play a critical role in CaOx stone formation across different populations. As requested, we have clearly incorporated a discussion of this limitation in the revised manuscript (Page 25).

Comments 32: [The conclusion that these genes are “potential therapeutic targets” seems premature. What evidence supports that modulating their expression could alter disease outcomes?]

Response 32: We thank the reviewer for this critical comment. We agree that directly labeling these genes as confirmed "therapeutic targets" overstates our current evidence, and we have therefore revised our conclusion throughout the manuscript to more accurately describe them as "assess their potential for therapeutic intervention." "candidate key targets regulating CaOx kidney stones". The rationale for their candidacy is as follow:

Independent validation: Their dysregulation was confirmed in an independent patient cohort (GSE73680) (Figure 4).

Pathogenic link: They are enriched in pathways like “olfactory transduction”, which in renal CaOx injury and stone disease is an emerging field (Figure 4).

Dynamic activity: Pseudotime analysis reveals these genes are dynamically expressed across endothelial cell state transitions (Figure 7).

We have clarified that this work is hypothesis-generating. Future functional studies (e.g., gene modulation in disease models) are essential to formally test if targeting these genes alters disease outcomes (Page 25).

Comments 33: [The limitations section briefly mentions small sample size and lack of functional validation. Shouldn’t the absence of independent replication and in vivo evidence be emphasized more explicitly?

Response 33: We thank the reviewer for this valuable suggestion. We agree that the absence of independent replication and in vivo functional evidence are critical limitations that warrant explicit emphasis. In direct response to this comment, we have now substantially revised the Limitations section to highlight these two points more explicitly and with greater prominence, stating that our findings, being derived from a single demographic cohort and lacking in vivo validation, require confirmation through future multi-center studies and direct experimental evidence to establish their generalizability and causal relevance (Page 25).

Comments 34: [Could the gene signatures identified reflect a general response to renal injury rather than specific to CaOx-induced stones?]

Response 34: We thank the reviewer for raising this critical point. We agree that distinguishing a general injury response from a CaOx-specific signature is challenging and that definitive proof of specificity requires causal validation, which we acknowledge as a limitation in our revised discussion (Page 25). However, we have employed several analytical strategies to strengthen the argument for their relevance to CaOx stone disease:

Disease-specific context: The genes were identified in Randall's plaque tissues (GSE176155, comprised 3 randall’s plaque and 3 normal kidney papillae tissues) (Figures 1-4), the specific precursor of CaOx stones (PMID: 33514941).

Independent validation: Their dysregulation was confirmed in the kidney tissue containing CaOx kidney stones (GSE73680, consisted of 29 samples of kidney tissue containing CaOx kidney stones and 33 control samples) (Figure 4).

Dynamic expression: Pseudotime analysis shows their dynamic regulation during endothelial remodeling in this specific context (Figure 7).

Therefore, while general injury involvement cannot be ruled out, our data robustly position these genes within the specific pathological continuum of CaOx stone disease.

Comments 35: [The predictive model lacks clinical variables (e.g., urinary oxalate, calcium, or kidney function). How might inclusion of such parameters alter model performance?]

Response 35: We thank the reviewer for this pertinent observation. We completely agree that the inclusion of key clinical parameters, such as urinary metabolites and kidney function, would likely enhance the predictive performance and clinical translatability of our model.

Unfortunately, as we have now explicitly acknowledged in the 'Limitations' section, the public dataset (GSE73680) used for model construction and validation lacks these detailed clinical variables. We fully endorse the reviewer's view and have proposed that the integration of comprehensive clinical data with transcriptomic profiles constitutes a critical and necessary step for future research (Page 25).

We once again extend our sincere gratitude for dedicating your valuable time to reviewing our manuscript and for your insightful suggestions. We have revised and improved the manuscript according to your comments, and your feedback has been instrumental in significantly enhancing its quality. We are very pleased to have incorporated your recommendations and appreciate your recognition of our work, which inspires us to continue our in-depth research. We look forward to your further guidance.

Round 2

Reviewer 2 Report

Comments and Suggestions for Authors

Thank you for improving the manuscript.

Reviewer 3 Report

Comments and Suggestions for Authors

The paper can be accepted in its present form.